# WHERE REASONING FAILS: STEP-WISE CONFIDENCE ATTRIBUTION IN BLACK-BOX LLMS

## ABSTRACT

Large Language Models (LLMs) have achieved strong performance on complex reasoning tasks by generating step-by-step solution traces, but diagnosing where a reasoning trace might fail remains difficult. Confidence estimation (CE) provides reliability signals but is usually restricted to the final answer, offering only coarse diagnostics. While recent studies have explored stepwise diagnostics, existing methods rely on white-box access, such as token-level logits or fine-tuned models, which are infeasible for closed-source LLMs. We introduce Stepwise Confidence Attribution, a black-box framework for diagnosing errors, requiring only access to generated reasoning traces. Stepwise confidence attribution applies the Information Bottleneck (IB) principle to assign confidence scores at the step level, treating consensus structures across correct solutions as anchors of reliable reasoning with high confidence. Steps that do not align with these consensus patterns are assigned lower confidence. We propose two complementary methods: (1) a non-parametric overlap-based approach (NIBS) that measures consistency without graph context, and (2) a Graph-based IB model (GIBS) that learns subgraphs through a differentiable mask to capture structural variability. Through extensive experiments on mathematical reasoning and multi-hop question answering, we show that our framework reliably identifies low-confidence steps strongly correlated with reasoning errors. Moreover, incorporating step-level CE improves overall reasoning accuracy, yielding up to an 12.3% accuracy gain. Our framework provides a practical diagnostic tool for enhancing the reliability of LLM reasoning. Code can be found in the `https://anonymous.4open.science/r/ICLR_2026_-2801`.

## 1 INTRODUCTION

The ability to diagnose where a reasoning trace fails is essential for improving the reliability of large language models (LLMs). Solution traces such as Chain-of-Thought (CoT) (Wei et al., 2022) or Graph-of-Thought (GoT) (Besta et al., 2024) provide transparency into model reasoning, yet intermediate errors remain hard to identify and can critically affect the final prediction. Recent work has explored step-level diagnostics for reasoning traces, which largely fall into two categories. The first trains supervised classifiers with step-by-step human annotations to label whether each reasoning step is correct (Jiao et al., 2025; Zheng et al., 2024). The second prompts the LLM itself to be a judge to critique each solution step by step (Weng et al., 2023; Li et al., 2024). While both directions can provide useful signals, the former requires expensive human annotation, and the latter inherits bias and inconsistency from the judge model, which limits scalability and reliability.

Confidence estimation (CE) provides a complementary direction for assessing reliability, as it does not rely on external supervision and can operate directly on model outputs. Prior work has shown that measures such as semantic variance across sampled generations (Lin et al., 2023) or predictive entropy over logits (Lin et al., 2024) provide informative signals for estimating whether a final answer is correct. However, restricting CE to the final answer yields only a coarse reliability signal and fails to indicate which specific step in a reasoning trace is responsible for an error. This limitation motivates the need for Stepwise Confidence Attribution (SCA), where the goal is to assign confidence scores to individual reasoning steps and thereby provide fine-grained diagnostic signals.

Extending confidence attribution from the final-answer level to the step-wise setting introduces a dual challenge. On the one hand, reasoning traces generated by LLMs exhibit substantial *output*

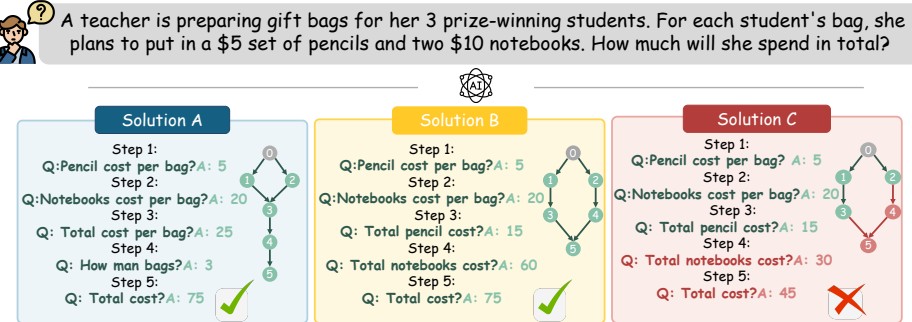

Figure 1: Example of reasoning trace variability in GSM8K dataset. Two distinct solution paths (A and B) yield the same correct answer, while another path (C) contains an erroneous step leading to a wrong result. Step-wise confidence attribution needs to distinguish legitimate variability from true structural inconsistencies.

*variability*: correct solutions may differ in step order, expression, or level of detail. On the other hand, many of these diverse traces still share a latent *common structure* that captures the essential reasoning path to the answer. A robust step-wise confidence estimator must distinguish harmless variability from true errors, where harmless variability reflects alternative but valid solution paths, whereas true errors are deviations that conflict with the shared reasoning structure. For example, in Figure 1, Solution A computes the per-bag cost before scaling by the number of students, whereas Solution B separately computes pencil and notebook totals before summing them. Although the intermediate steps differ, both solutions are consistent with the same underlying reasoning pattern and should therefore receive high confidence scores in the steps of both reasoning paths. In contrast, if a step introduced an incorrect operation, such as multiplying the wrong quantities as in Solution C, this deviation would undermine the common structure and justifiably be assigned low confidence. The key challenge is to maintain high confidence for legitimate variability while flagging only those steps that represent true structural inconsistencies.

Our key idea is to aggregate multiple solution traces and identify common steps that consistently appear across correct answers. These common steps serve as anchors of reliable reasoning and are assigned high confidence, while steps absent from consensus patterns are assigned low confidence, since they are more likely to reflect spurious or error-prone reasoning. Consensus structures thus act as a proxy for the latent reasoning pattern of a problem, enabling confidence attribution that is both fine-grained and robust.

This intuition of finding a shared structure amidst noisy variations maps naturally onto the **Information Bottleneck (IB)** principle. IB provides a formal language for balancing two competing objectives: compressing the input trajectory by discarding non-essential variations (the compression term), while retaining maximal information about the underlying correct reasoning pattern (the relevance term). Here, the input $X$ is a reasoning trajectory composed of multiple steps, the compressed representation $Z$ is the trajectory expressed with confidence weights on its steps, and the target $Y$ denotes the correctness signal of the trajectory. The objective is $\min_Z I(X; Z) - \beta I(Z; Y)$, where the first term encourages compression by selecting only a sparse subset of steps, and the second term ensures that this subset retains critical information required for correctness. Within this framework, we explore two complementary instantiations:

- **Non-parametric IB for Step-wise Confidence (NIBS).** $Z$ is realized as a set of consensus steps derived from correct solutions, and step-level confidence is obtained by measuring how well a trajectory aligns with this set.

- **Graph IB for Step-wise Confidence (GIBS).** To better handle structural variability, trajectories are represented as graphs, and $Z$ is a subgraph selected through a differentiable mask. Confidence scores are produced by aligning the selected subgraph with correctness signals, providing a more flexible treatment of structural variability.

Empirical results show that both IB-based methods produce accurate confidence estimates. Beyond diagnostics, we demonstrate the impact of fine-grained CE on downstream LLM performance, where using step-level signals for selective correction improves final-answer accuracy by up to 12.3% over state-of-the-art methods. Additional ablation studies verify the contribution of each component, and out-of-distribution (OOD) evaluations confirm that our framework generalizes well to reasoning traces outside the training domain.

## 2 RELATED WORK

**Reasoning Verification in LLMs.** Reasoning verification studies whether intermediate steps or final answers in multi-step reasoning are correct. Depending on the level of granularity, existing approaches can be broadly categorized into answer-level verification and step-level verification. **Answer-level methods** include self-consistency voting (Wang et al., 2022), reranking with outcome-based verifiers (Uesato et al., 2022; Zhang et al., 2024), and LLM-as-a-judge paradigms for scoring or pairwise comparison (Li et al., 2024). While these approaches have shown strong empirical performance, they often overlook the reasoning process itself, making it difficult to diagnose where errors originate (Tyen et al., 2023). **Step-level methods**, in contrast, trains models or verifiers to judge the correctness of intermediate steps, either through human-labeled supervision (Lightman et al., 2023a; Zheng et al., 2024), automatically constructed process reward models (Wang et al., 2023a; Setlur et al., 2024), or graph-structured verification (Cao, 2023; Fang et al., 2025; Mukherjee et al., 2025). Dedicated benchmarks such as REVEAL (Jacovi et al., 2024) further highlight the value of step-level diagnosis. However, existing approaches either rely on costly human annotations (Lightman et al., 2023b) or on subjective model judgments that may be unreliable (Szymanski et al., 2024; Stechly et al., 2024). In contrast, our work introduces quantitative, black-box confidence signals for each intermediate step, thereby reducing annotation costs and mitigating bias from model self-evaluation.

**Confidence Estimation in LLMs.** Uncertainty quantification (UQ) captures distribution-level variability (Gal & Ghahramani, 2016; Malinin & Gales, 2018), but offers only a coarse reliability notion. Confidence estimation (CE) instead operates at the sample level, assigning reliability scores to individual outputs (Lin et al., 2023). Most CE methods focus on the final answer, either through internal signals such as token-level logits and entropy (Kuhn et al., 2023; Lin et al., 2024), or through black-box signals such as agreement across sampled outputs (Lin et al., 2023). However, these methods produce only global scores and give no insight into intermediate steps. Recent step-wise CE approaches, such as CoT Entropy (Ye et al., 2025) and FineCE (Han et al., 2025), attempt to assign confidence along reasoning traces, but they require white-box access to token probabilities, restricting them to open-source models. Some works also model reasoning as graphs (Besta et al., 2024; Pandey et al., 2025), but the representation itself is orthogonal to our problem setting. Our work differs by introducing the problem of step-wise confidence attribution in the black-box setting, which provides quantitative and scalable confidence signals at the step level using only generated reasoning traces and correctness labels.

## 3 PROBLEM STATEMENT

We begin with the notion of answer-level CE. Let an LLM be represented as a probabilistic model $\mathcal{M}$ that generates a response $y$ conditioned on input $x$. CE assigns a reliability score to the final answer $A$. For open-source models, confidence can be defined from token probabilities, $C(x, A) = p(A|x; \mathcal{M})$. In black-box settings, however, token-level probabilities are unavailable, so confidence must instead be inferred from agreement among $N$ sampled outputs $\{A_1, A_2, \ldots, A_N\}$.

**Problem 1 (Answer-level CE)** *Given an input $x$ and $N$ sampled answers $\{A_i\}_{i=1}^N$, the goal of answer-level CE is to learn a mapping*

$$f : \{A_i\}_{i=1}^N \to \{c_i\}_{i=1}^N,$$

*where $c_i$ is the confidence score of answer $A_i$, estimated from observable signals such as agreement or semantic similarity among the sampled outputs.*

While effective, answer-level confidence estimation (CE) provides only a coarse reliability signal tied to the final output, without revealing *which reasoning steps* contribute to success or failure. This limitation becomes especially problematic in high-stakes decision-making, model debugging, or interactive human–AI collaboration, where identifying the precise locus of failure is critical. To address this, we shift the granularity of CE from the final answer to the intermediate reasoning process. In reasoning models (Guo et al., 2025), each output is not a single answer but a **reasoning trajectory** $y_i = (T_i, A_i)$, where $T_i = \{t_{i1}, t_{i2}, \ldots, t_{iL_i}\}$ is a sequence of intermediate steps and $A_i$ is the final answer. Generic step-wise CE methods aim to assign reliability scores to intermediate steps, sometimes without requiring correctness labels. In this work, however, we focus on a more specific setting, which we call **Step-wise Confidence Attribution (SCA)**. In SCA, final-answer correctness labels are available and used to construct consensus anchors that guide step-level scoring.

**Problem 2 (Step-wise Confidence Attribution)** *Given an input $x$ and $N$ sampled trajectories $\mathcal{S} = \{(T_i, A_i, z_i)\}_{i=1}^N$ with correctness labels $z_i \in \{0, 1\}$, the goal is to learn a mapping*

$$f : (T_i, \mathcal{S}) \rightarrow \{c_{ij}\}_{j=1}^{L_i},$$

*where $c_{ij}$ is the confidence score assigned to step $t_{ij}$ in $T_i$, reflecting its alignment with common structures across correct trajectories.*

The primary goal of SCA is to serve as a diagnostic tool, attributing confidence scores to steps based on their alignment with a consensus of correct reasoning. For an incorrect trajectory ($z_i = 0$), $f$ should assign low confidence to the inconsistent steps most likely responsible for the error. For a correct trajectory ($z_i = 1$), not all steps need uniformly high confidence; rather, steps that match the consensus among correct solutions should receive higher confidence, while unusual steps may be assigned lower scores to capture structural or semantic variability. While our formulation assumes access to final-answer correctness labels, this assumption is natural for the task itself: identifying *where reasoning goes wrong* requires distinguishing correct from incorrect solutions. For LLM evaluation/diagnosis (Liu et al., 2023; Augenstein et al., 2024; Gao et al., 2025; Zhao et al., 2024; Wang et al., 2023b), golden final answers are typically available for benchmarking. Correctness provides a lightweight supervision signal that avoids reliance on token-level logits, additional LLM judges, or costly step-level human annotations, making our setting both practical and well aligned with the goal of scalable step-wise confidence attribution.

## 4 METHOD

Extending confidence estimation to the step-wise setting introduces the key challenge of distinguishing benign output variability from true reasoning errors. To tackle this challenge, our approach is rooted in a key insight: while individual correct solutions may vary on the surface, they often share a latent common structure that reflects the essential reasoning pattern. A robust attribution method must therefore learn to identify this shared structure and assign confidence based on a step-level consistency with it. Our method employs final-answer correctness labels, a lightweight yet powerful source of supervision, to discover these latent structures by constructing consensus anchors from correct reasoning traces. We leverage these consensus anchors to guide confidence attribution, formalizing the problem under the **Information Bottleneck (IB)** principle.

### 4.1 INFORMATION BOTTLENECK FORMULATION

We cast step-wise confidence attribution as an instance of the Information Bottleneck (IB) principle. Given a trajectory $T_i = \{t_{i1}, \ldots, t_{iL_i}\}$ sampled from the LLM, the goal is to produce a confidence mask $Z = \{c_{ij}\}_{j=1}^{L_i}$ over its steps. Unlike settings with step-level annotations, we do not observe the correctness of individual steps. Instead, we only have access to final-answer labels $z_i \in \{0, 1\}$ for each trajectory. The target $Y$ must therefore be derived from the sampled set $\mathcal{S} = \{(T_i, A_i, z_i)\}$. Concretely, $z_i$ partitions $\mathcal{S}$ into correct and incorrect subsets, and consensus anchors $\mathbf{m}_{ij}$ are aggregated from $\mathcal{S}_{\text{correct}}$ to approximate the latent reasoning structure. The IB objective is

$$\min_Z \; I(T_i; Z) - \beta I(Z; Y), \tag{1}$$

where $I(T_i; Z)$ encourages compression by sparsely selecting steps, and $I(Z; Y)$ ensures that the retained steps align with correctness signals inferred from consensus. In the following, we instantiate this principle with two variants: (1) **NIBS**, a non-parametric overlap-based approach, and (2) **GIBS**, a graph-based model that applies a differentiable IB mask to handle structural variability.

### 4.2 NON-PARAMETRIC IB FOR STEP-WISE CONFIDENCE (NIBS)

Directly solving the IB objective in Eq. 1 is generally intractable, since it requires searching over all possible confidence values of steps in $T_i$ and estimating mutual information terms. A natural approximation is to assume that steps consistently appearing in correct trajectories are the most informative about correctness $Y$, while steps absent from correct trajectories carry little predictive value. Under this approximation, the IB objective reduces to retaining consensus steps as the compressed representation $Z$, and assigning higher confidence to steps that align with this consensus. Formally, given a trajectory $T_i = \{t_{i1}, \ldots, t_{iL_i}\}$ and one of its steps $t_{ij}$, its confidence can be computed as

$$c_{ij} \; = \; \mathbb{E}_{S \sim \mathcal{S}_{\text{correct}}} \Big[ \text{Agg}\Big( \{\text{sim}(\mathbf{t}_{ij}, \mathbf{t}') \mid \mathbf{t}' \in S\} \Big) \Big], \tag{2}$$

where $\mathrm{sim}(\cdot, \cdot)$ measures semantic similarity between steps (e.g., cosine (Golovneva et al., 2022) or NLI), and $\mathrm{Agg}$ aggregates similarities within a trajectory (e.g., maximum or mean).

This construction instantiates the IB principle: compression $I(T_i; Z)$ is realized by restricting $Z$ to consensus steps, while relevance $I(Z; Y)$ is maximized because consensus overlap is strongly correlated with correctness. NIBS provides a closed-form, non-parametric solution to the IB objective without requiring model training. However, NIBS ignores structural dependencies by matching steps based on the semantic similarity alone, potentially grouping nodes that are structurally disparate. This limitation motivates a graph-based IB formulation with learned subgraph selection.

## 4.3 GRAPH IB FOR STEP-WISE CONFIDENCE (GIBS)

While NIBS provides a simple closed-form instantiation of the IB principle, it ignores structural dependencies among reasoning steps. To capture such dependencies, we adopt a graph-based formulation in which each trajectory $T_i$ is represented as a directed graph $G_i = (V_i, E_i)$, where nodes denote intermediate results and edges encode logical or semantic dependencies, each reasoning step $t_{ij}$ is represented as a pair $(v_{ij}, e_{ij})$, where $v_{ij}$ is the intermediate result (node) and $e_{ij}$ is the edge that describes the reasoning operation producing it. In general, such graphs can be constructed either by prompting the LLM to output explicit dependency information, or by applying post-processing heuristics such as semantic dependency parsing, as done in prior work (Da et al., 2025; Chen et al., 2023; Amini et al., 2019). In this work, we adopt the former approach and elicit explicit dependency structures directly from the LLM outputs (see Appendix A.4.4 for the prompt). This representation allows step selection to account not only for surface similarity but also for the topology of reasoning. In this setting, $Z$ is instantiated as a selected subgraph $G^* \subseteq G_i$. Then the objective of Eq. 1 becomes

$$\min_{G^*} \ I(G_i; G^*) - \beta I(G^*; Y). \tag{3}$$

Since step-level labels are not observable, we approximate $Y$ by a consensus graph $G^{MC}$ aggregated from correct trajectories in the sampled set $\mathcal{S}$. For each reasoning graph $G_i$, we compute the Maximum Common Subgraph (MCS) (McCreesh et al., 2017) between $G_i$ and each correct graph $G_k \in \mathcal{G}_{\mathrm{correct}}$. Then, each MCS highlights the reasoning components shared between $G_i$ and a correct solution. Aggregating these pairwise MCS results yields $G^{MC}$, which serves as a consensus structure reflecting how $G_i$ aligns with correct reasoning. Details of calculating MCS and their aggregation can be found in Appendix A.2. Replacing $I(G^*; Y)$ with $I(G^*; G^{MC})$, the structure-alignment IB objective now becomes:

$$\min_{G^*} \ I(G_i; G^*) - \beta I(G^*; G^{MC}). \tag{4}$$

**From IB to Soft-mask Relaxation.** Directly solving the IB objective on graphs is intractable: selecting a discrete subgraph $G^* \subseteq G_i$ is a combinatorial problem, and estimating mutual information terms between discrete subgraphs and correctness signals is not feasible in practice. We therefore introduce a differentiable mask $\mathbf{p}_\theta = \{p_{\theta, ij}\}$ over steps, yielding a soft subgraph $G^* = G_i \odot \mathbf{p}$ as the compressed representation. For each step $t_{ij}$, the model predicts a selection probability $p_{ij} \in [0, 1]$ that reflects its contribution to the retained reasoning structure.

**Approximating the IB Objective.** The two mutual information terms in the IB objective are approximated with tractable surrogates:

- *Compression.* We start from the mutual information identity $I(G_i; G^*) = H(G^*) - H(G^* \mid G_i)$. In our formulation, the subgraph $G^*$ is deterministically obtained from $G_i$ given the mask $\mathbf{p}_\theta$, so the conditional entropy $H(G^* \mid G_i)$ vanishes. Thus, $I(G_i; G^*) \approx H(G^*)$. Under the independent Bernoulli masking assumption, the entropy of the subgraph is equivalent to the entropy of the mask distribution: $H(G^*) \approx H(\mathbf{p}_\theta) = -\sum_j \left[ p_{\theta, ij} \log p_{\theta, ij} + (1 - p_{\theta, ij}) \log(1 - p_{\theta, ij}) \right]$. This term penalizes diffuse selections and encourages sparsity in the chosen subgraph. While some formulations of IB maximize entropy to learn a compressed stochastic representation, our goal is to select a single, determinate subgraph. By minimizing the entropy of the mask distribution, we force the model to make confident, binary-like decisions for each step (i.e., $p_{\theta, ij} \to 0$ or 1). This directly encourages a compressed representation of the reasoning graph, thus satisfying the compression objective $I(G_i; G^*)$ in a hard, non-variational manner.

- *Relevance.* Similarly, we can decompose $I(G^*; Y) = H(Y) - H(Y \mid G^*)$. Since $H(Y)$ is constant, maximizing $I(G^*; Y)$ is equivalent to minimizing the conditional entropy $H(Y \mid G^*)$. Direct estimation of this term is infeasible without step-level labels. Instead, we approximate

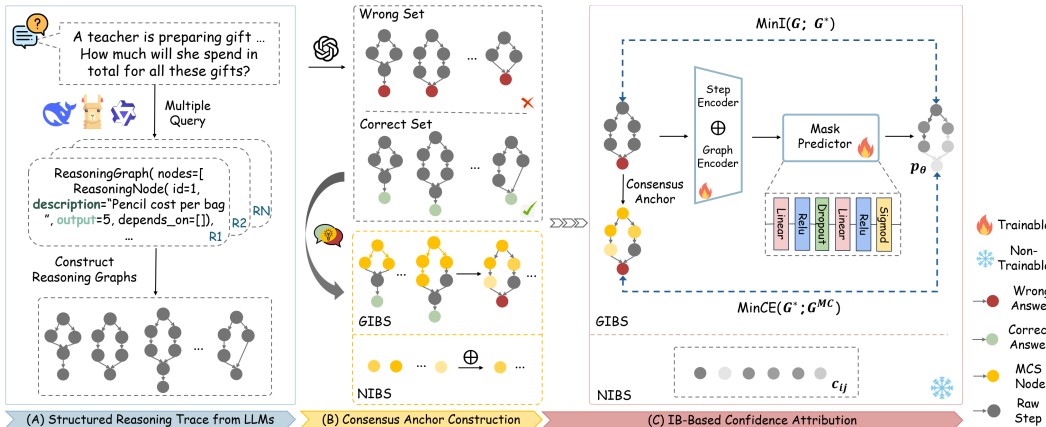

Figure 2: Overview of the IB-based step-wise confidence attribution framework. The process consists of (A) constructing structured reasoning traces from LLM outputs, (B) deriving consensus anchors from correct trajectories, and (C) applying the IB formulation through NIBS and GIBS to produce confidence scores.

supervision through a consensus graph $G^{MC}$ aggregated from correct trajectories in $\mathcal{S}_{\text{correct}}$. By aligning $G^*$ with this consensus structure, we approximate the conditional entropy term via cross-entropy between the predicted mask $\mathbf{p}_\theta$ and the consensus mask $\mathbf{m}_i$: $I(G^*; Y) \gtrsim -\text{CE}(\mathbf{p}_\theta, \mathbf{m}_i)$, where each $m_{ij} \in \{0, 1\}$ indicates whether step $t_{ij}$ belongs to the consensus reasoning structure. This surrogate encourages selected steps to align with reliable reasoning patterns observed in correct solutions.

**Training Objective.** For each reasoning graph $G_i$, the model $f_\theta$ outputs a soft mask $\mathbf{p}_\theta = f_\theta(G_i)$, where $p_{\theta,ij} \in [0, 1]$ denotes the selection probability for step $t_{ij} = (v_{ij}, e_{ij})$. The consensus mask $\mathbf{m}_i$ is obtained by aligning $G_i$ with the maximum common subgraph $G^{MC}$ constructed from correct trajectories. Each $m_{ij} \in \{0, 1\}$ indicates whether step $t_{ij}$ is part of the consensus reasoning structure. The final loss is

$$\mathcal{L}(G_i) = H(\mathbf{p}_\theta) + \lambda \, \text{CE}(\mathbf{p}_\theta, \mathbf{m}_i). \tag{5}$$

**Inference.** At test time, given a new reasoning graph $G$ without any gold final-answer or step-level correctness labels, the model outputs a probability mask $\mathbf{p}_\theta = f_\theta(G)$. For each step $t_{ij} = (v_{ij}, e_{ij})$, the step-wise confidence score is $c_{ij} = p_{\theta,ij}$. Steps with high probabilities are considered reliable, while those with low probabilities are flagged as structurally inconsistent or error-prone.

## 4.4 IMPLEMENTATION

Figure 2 illustrates the overall pipeline when we implement our step-wise confidence attribution under the IB formulation. The detailed algorithms can be found in the Appendix A.2 and complexity analysis in the Appendix A.3. **(A) Structured Reasoning Trace from LLMs.** Given a problem, we query an LLM multiple times to obtain diverse solution traces. Each trace is parsed into a reasoning graph $G_i = (V_i, E_i)$, where nodes store intermediate results and edges encode the reasoning operations. Each step is represented as a node–edge pair $(v_{ij}, e_{ij})$, where $v_{ij}$ is the result and $e_{ij}$ is the operation producing it (Da et al., 2025; Chen et al., 2023; Amini et al., 2019). In practice, we elicit this structure via a LangFun-style[1]. The full prompt we use is provided in Appendix A.4.4. Then, we extract the reasoning graph using a simple rule-based parser. A concrete example is shown in our case study in Figure 1. **(B) Consensus Anchor Construction.** Based on the correctness of the final answer, we partition trajectories into correct and wrong sets, where correctness is judged against gold answers (e.g., by GPT-4o). Consensus anchors are constructed only from the correct set, approximating latent step-level labels. In NIBS, each step is compared to steps in the correct set via a similarity function. In GIBS, consensus anchors are obtained by maximum common subgraph (MCS) (McCreesh et al., 2017) matches between candidate and correct graphs, and steps that consistently appear in the MCS are treated as high-confidence. **(C) IB-Based Confidence Attribution.** We apply the IB principle to assign step-wise confidence scores. NIBS directly computes confidence from consensus anchors in closed form and is fully non-trainable. In contrast, GIBS uses a trainable mask predictor: step and graph encoders produce contextualized representations, which are passed into the predictor to output soft selection probabilities $p_\theta$.

---

[1] https://github.com/google/langfun

## 5 EXPERIMENTS

We conduct experiments to evaluate the effectiveness of our step-wise confidence attribution methods. Specifically, we address the following research questions:

• **RQ1 (Accuracy)**: How accurately can our proposed methods (NIBS and GIBS) identify erroneous steps in a reasoning trace compared to strong baselines?

• **RQ2 (Utility)**: Can the step-wise confidence scores from our methods be used to improve the final-answer accuracy of LLMs through targeted self-correction?

• **RQ3 (Ablation & Generalization)**: What is the contribution of key components in our framework How well does our learned model (GIBS) generalize to out-of-distribution data?

### 5.1 EXPERIMENT SETTINGS

Below we describe the datasets, compared methods, configurations, and evaluation metrics used. Additional implementation details and prompt templates are provided in Appendix A.4.

**Datasets.** We evaluate on three reasoning benchmarks: (1) **GSM8K** (Cobbe et al., 2021), a widely used math word problem dataset; (2) **Math** (Hendrycks et al., 2021), competition-level problems requiring more complex reasoning; and (3) **MoreHopQA** (Schnitzler et al., 2024), a multi-hop QA dataset testing generalization beyond mathematics.

**Compared Methods.** We compare the following methods: (i) four *white-box* CE approaches adapted to step-wise reasoning: SL(norm) (Lin et al., 2024; Cole et al., 2023), Token Entropy (Kuhn et al., 2023), P(true) (Kadavath et al., 2022), and LeCo (Yao et al., 2024); (ii) our proposed **NIBS** family, which instantiates the closed-form IB solution with different similarity functions (cosine, NLI) and aggregation strategies (mean, max, random); and (iii) our **GIBS** model, which learns to align subgraphs with consensus structures.

**Configurations.**

We evaluate three representative LLMs: LLaMA-3.1-8B-Instruct (Grattafiori et al., 2024), Phi-4-Reasoning (Abdin et al., 2025), and DeepSeek-R1-Distill-Qwen-32B (Guo et al., 2025). For each input, we sample $N = 20$ traces with temperature 1.0. NIBS uses cosine similarity over sentence embeddings from `bert-base uncased` (Devlin et al., 2019), and both NIBS and GIBS use NLI-based similarity from the `DeBERTa-large-MNLI` model (He et al., 2020; Lin et al., 2023). GIBS is trained on 2,000 reasoning graphs constructed from sampled solutions. All results are averaged over 10,000 trajectories per dataset. **Metrics.** Following prior work in CE and calibration (Davis & Goadrich, 2006; Lin et al., 2023; Geifman & El-Yaniv, 2017), we adopt four complementary metrics: (i) **AUROC** and (ii) **AUCPR** evaluate ranking quality of confidence scores, with AUCPR being especially important under class imbalance since erroneous steps are sparse. (iii) **ACC@80%** measures selective prediction performance, i.e., the final-answer accuracy when retaining only the top 80% most confident steps, reflecting utility in downstream self-correction. (iv) **ECE** (Expected Calibration Error) assesses calibration by comparing predicted confidence with empirical accuracy across bins.

### 5.2 ACCURACY OF STEP-WISE CONFIDENCE ATTRIBUTION (RQ1)

We begin by validating the core assumption of our consensus-based framework: correct reasoning trajectories should exhibit stronger overlap in their common substructures. Figure 3 shows the distribution of average maximum common subgraph (MCS) scores across 1,000 reasoning graphs. Correct solutions concentrate around larger relative MCS sizes, while incorrect solutions peak around smaller values. This suggests that correctness is associated with stability and reproducibility of the reasoning path, while erroneous traces tend to diverge, producing fragmented structures. These observations justify the use of consensus as a proxy supervision signal for step-wise CE.

We then compare NIBS and GIBS against strong white-box baselines for step-wise CE. Results in Table 1 show that GIBS achieves the best overall performance across models and datasets. The gain comes from explicitly modeling structural dependencies: by aligning subgraphs with consensus anchors, GIBS suppresses incidental variability while emphasizing steps that break the shared reasoning structure. On GSM8K and Math, GIBS outperforms NIBS variants (e.g., Cos-Mean, NLI-Mean), which only capture local semantic overlap but ignore dependencies. On MoreHopQA, GIBS achieves the best AUROC (0.6619) and AUCPR (0.6866), showing that structure-aware selection helps when reasoning spans multiple passages.

| LLM | Llama3.1-8b | | | | DeepSeek-R1-Distill-Qwen-32B | | | | Phi4-reasoning | | | |
|---|---|---|---|---|---|---|---|---|---|---|---|---|
| Metrics | AUROC ↑ | AUCPR ↑ | ACC@80% ↑ | ECE ↓ | AUROC ↑ | AUCPR ↑ | ACC@80% ↑ | ECE ↓ | AUROC ↑ | AUCPR ↑ | ACC@80% ↑ | ECE ↓ |
| Dataset: GSM8K | | | | | | | | | | | | |
| P(true) | 0.4016 | 0.4711 | 0.5283 | 0.5504 | 0.5159 | 0.5820 | 0.5840 | 0.5802 | 0.5251 | 0.6956 | 0.6934 | 0.6851 |
| SL(norm) | 0.4790 | 0.5240 | 0.5513 | 0.2282 | 0.3700 | 0.5308 | 0.5262 | 0.1230 | 0.3851 | 0.5947 | 0.6799 | 0.2394 |
| Entropy | 0.4105 | 0.4962 | 0.5141 | 0.2518 | 0.5203 | 0.5550 | 0.5235 | 0.2583 | 0.4623 | 0.6795 | 0.6664 | 0.1648 |
| LECO | 0.3862 | 0.4586 | 0.5319 | 0.3783 | 0.3202 | 0.4110 | 0.4883 | 0.3395 | 0.2885 | 0.5735 | 0.6344 | 0.4021 |
| Cos-Max | 0.4537 | 0.5152 | 0.5513 | 0.3780 | 0.5269 | 0.5197 | 0.5676 | 0.3799 | 0.3494 | 0.5861 | 0.7010 | 0.1997 |
| Cos-Mean | 0.6078 | 0.6211 | 0.6175 | 0.3300 | 0.6633 | 0.6933 | 0.5748 | 0.3323 | 0.5959 | 0.7703 | 0.7199 | 0.1556 |
| NLI-Max | 0.7096 | 0.7890 | 0.5908 | 0.1162 | 0.7450 | 0.6982 | 0.6409 | 0.1456 | 0.6600 | 0.8141 | 0.7446 | 0.2009 |
| NLI-Mean | 0.5524 | 0.6103 | 0.5665 | 0.4376 | 0.6762 | 0.6508 | 0.6318 | 0.4189 | 0.5738 | 0.7704 | 0.7186 | 0.5527 |
| GIBS | 0.6910 | 0.7004 | 0.6292 | 0.2293 | 0.7289 | 0.6712 | 0.6532 | 0.2867 | 0.7892 | 0.8172 | 0.8117 | 0.3354 |
| Dataset: MoreHopQA | | | | | | | | | | | | |
| P(true) | 0.5228 | 0.5486 | 0.5450 | 0.5357 | 0.5177 | 0.6492 | 0.6606 | 0.6443 | 0.5086 | 0.6159 | 0.6362 | 0.6203 |
| SL(norm) | 0.4005 | 0.4506 | 0.5187 | 0.3168 | 0.2463 | 0.4894 | 0.6049 | 0.2829 | 0.3198 | 0.4966 | 0.5891 | 0.2751 |
| Entropy | 0.5510 | 0.5413 | 0.5586 | 0.2148 | 0.6103 | 0.7150 | 0.6812 | 0.0952 | 0.6012 | 0.6702 | 0.5709 | 0.0520 |
| LECO | 0.3280 | 0.4365 | 0.4921 | 0.3501 | 0.3116 | 0.5777 | 0.5642 | 0.3700 | 0.2760 | 0.4944 | 0.5782 | 0.4254 |
| Cos-Max | 0.3836 | 0.4446 | 0.5179 | 0.3687 | 0.3390 | 0.5129 | 0.6031 | 0.3066 | 0.3454 | 0.5284 | 0.6413 | 0.2614 |
| Cos-Mean | 0.5044 | 0.5487 | 0.5275 | 0.3160 | 0.5938 | 0.6919 | 0.6419 | 0.2434 | 0.4779 | 0.6492 | 0.6388 | 0.2023 |
| NLI-Max | 0.4937 | 0.5485 | 0.5347 | 0.2159 | 0.6663 | 0.7766 | 0.6457 | 0.1107 | 0.5801 | 0.6637 | 0.6961 | 0.2323 |
| NLI-Mean | 0.5124 | 0.5535 | 0.5314 | 0.3902 | 0.6291 | 0.6899 | 0.6498 | 0.4305 | 0.5440 | 0.6428 | 0.6776 | 0.4610 |
| GIBS | 0.6471 | 0.6694 | 0.5602 | 0.3173 | 0.8084 | 0.8357 | 0.7051 | 0.1832 | 0.6619 | 0.6866 | 0.7053 | 0.3560 |
| Dataset: Math | | | | | | | | | | | | |
| P(true) | 0.4584 | 0.4166 | 0.4551 | 0.4564 | 0.5055 | 0.6298 | 0.6226 | 0.6215 | 0.5277 | 0.7435 | 0.7554 | 0.7416 |
| SL(norm) | 0.5138 | 0.4627 | 0.4626 | 0.3250 | 0.4093 | 0.5433 | 0.6089 | 0.2319 | 0.3841 | 0.6597 | 0.7340 | 0.2220 |
| Entropy | 0.5120 | 0.4617 | 0.4550 | 0.2876 | 0.5190 | 0.6259 | 0.6225 | 0.1814 | 0.5190 | 0.7351 | 0.7479 | 0.2023 |
| LECO | 0.4378 | 0.4252 | 0.4382 | 0.2229 | 0.3838 | 0.5836 | 0.5901 | 0.1813 | 0.4089 | 0.7279 | 0.7044 | 0.3957 |
| Cos-Max | 0.4313 | 0.4021 | 0.4513 | 0.4408 | 0.3376 | 0.5015 | 0.5937 | 0.3093 | 0.3845 | 0.6617 | 0.7656 | 0.1568 |
| Cos-Mean | 0.5118 | 0.4848 | 0.4607 | 0.3946 | 0.5363 | 0.6583 | 0.6177 | 0.2463 | 0.6116 | 0.8503 | 0.7708 | 0.1013 |
| NLI-Max | 0.5173 | 0.5061 | 0.4743 | 0.1852 | 0.5310 | 0.6676 | 0.6340 | 0.2340 | 0.6043 | 0.8052 | 0.7925 | 0.3090 |
| NLI-Mean | 0.5148 | 0.4881 | 0.4670 | 0.3356 | 0.5407 | 0.6694 | 0.6267 | 0.4950 | 0.5739 | 0.8020 | 0.7712 | 0.5935 |
| GIBS | 0.5855 | 0.4890 | 0.4513 | 0.2737 | 0.5806 | 0.6831 | 0.6359 | 0.3786 | 0.6946 | 0.8078 | 0.8322 | 0.4050 |

Table 1: Overall results for step-wise confidence attribution on the GSM8K, MoreHopQA, and Math datasets. Our proposed method, GIBS, consistently outperforms baseline methods, especially on more complex reasoning tasks. The best and second-best results for each metric are marked in bold and with an underline, respectively. Higher AUROC, AUCPR, and ACC@80% and lower ECE indicate better performance.

Among the NIBS variants, NLI-based methods (especially NLI-Max) perform competitively on GSM8K and Math because semantic entailment captures local correctness, though performance degrades on MoreHopQA where structural alignment across passages is essential. White-box token-level methods such as Entropy achieve low calibration error (ECE = 0.0520 on MoreHopQA), but without structural modeling, they struggle to rank erroneous steps reliably. Across all datasets, Math proves most challenging, yet GIBS maintains the highest AU-ROC (0.6946) and competitive AUCPR (0.8078), demonstrating robustness to complex reasoning.

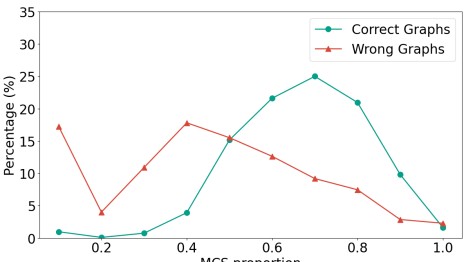

Figure 3: Distribution of average MCS proportion over 1,000 reasoning graphs. Correct graphs (green) concentrate near 0.8, while incorrect ones (red) peak near 0.4. This indicates that correct solutions are likely to follow a stable reasoning path.

## 5.3 Leveraging Step-wise Confidence for Error Correction (RQ2)

Previous experiments show we can detect errors, in this section, we show that this detection can actually be used to improve accuracy. Specifically, we study whether highlighting low-confidence steps helps the model revise erroneous reasoning. Our evaluation considers MoreHopQA instances that the LLM initially answered incorrectly, and asks LLMs to regenerate with two feedback: (1) **Final-answer feedback**: the model is only told that its final answer was wrong. (2) **Step-wise feedback**: in addition to answer-level feedback, the model is shown its previous reasoning trace with low-confidence steps explicitly marked. Figure 4 shows that step-wise feedback leads to substantially higher correction rates than answer-level feedback alone. The improvement is most pronounced for stronger models such as DeepSeek-R1 and Phi-4-Reasoning,

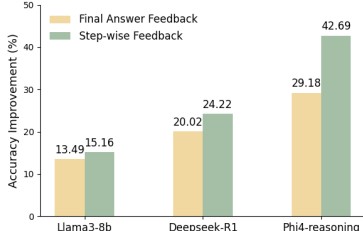

Figure 4: Effect of step-level feedback on correcting initially wrong answers in MoreHopQA. The baseline (yellow) provides only answer-level feedback, while our method (green) also highlights low-confidence steps.

which can better exploit localized error signals to revise their reasoning. These results demonstrate that fine-grained CE not only identifies unreliable steps but also provides actionable guidance that boosts downstream accuracy.

We also visualize a case study in Appendix A.5, illustrating how providing such reasoning feedback enables an LLM to self-correct its initial error. Since early mistakes can propagate and compound

along the reasoning chain, we further report the performance of detecting the *first* error in Appendix A.6.

## 5.4 ABLATION STUDY

To investigate the effectiveness of different design choices in our framework, we conduct the following ablation experiments on Phi4-Reasoning across three datasets:

**(1) GIBS w/ vs. w/o edge or graph encoder.** The mask predictor in GIBS receives input from both an edge encoder, which captures local structural information, and a graph encoder, which provides global contextual information. From the results in Table 2, we can see that removing either component leads to a noticeable drop in AUROC, confirming that both local and global signals are essential for accurate step-wise confidence prediction.

| Method | GSM8K | MorehopQA | Math |
|---|---|---|---|
| GIBS | 0.7892 | 0.6619 | 0.6946 |
| w/o Graph Encoder | 0.7228 | 0.6481 | 0.5961 |
| w/o Edge Encoder | 0.5186 | 0.3760 | 0.4763 |

Table 2: Performance of GIBS w.r.t. AUROC with and without the edge encoder or the graph encoder.

**(2) MCS-only supervision vs. consensus regularization.** Our loss encourages the selected subgraph to align with the consensus among correct solutions, approximating MCS supervision. At inference time, however, calculating MCS is computationally extensive, and high-quality sets of correct solutions are often unavailable in practice. We therefore train GIBS with soft consensus regularization. As shown in Table 3, GIBS achieves performance comparable to explicit MCS supervision while reducing inference time by more than three orders of magnitude. This demonstrates that our method effectively approximates consensus structures without requiring costly MCS computation.

| Method | GSM8K | | MoreHopQA | | Math | |
|---|---|---|---|---|---|---|
| | AUROC ↑ | Time ↓ | AUROC ↑ | Time ↓ | AUROC ↑ | Time ↓ |
| GIB-based | **0.7892** | **3min** | **0.6619** | **1min** | 0.6946 | **20s** |
| MCS-Only | 0.7743 | 19h | 0.6593 | 10h | 0.7254 | 5h |

Table 3: Comparison of GIB-based and MCS-Only methods w.r.t AUROC and inference time.

## 5.5 GENERALIZATION AND ROBUSTNESS ANALYSIS

We evaluate the generalization capabilities of our framework from three perspectives: (1) generalization across domains (OOD), (2) robustness when final answer ground-truth labels are unavailable (weak supervision), and (3) adaptability to free-form reasoning formats.

### 5.5.1 CROSS-DOMAIN GENERALIZATION (OOD) PERFORMANCE

A practical CE method should remain effective beyond the domain it was trained on. To test this, we train GIBS with the trajectories generated by Phi4-Reasoning on MoreHopQA, a textual multi-hop QA dataset, as an in-distribution (ID) domain and evaluate it directly on the Math dataset without re-training as an out-of-distribution (OOD) domain. As shown in Figure 5, GIBS consistently achieves higher AUROC than NIBS and white-box baselines despite the domain shift. This advantage arises because NIBS relies on step similarity within the training domain, whereas GIBS learns structural representations under the IB principle, enabling it to capture abstract reasoning patterns that are less tied to domain-specific vocabulary. These results suggest that graph-based modeling provides strong robustness to domain variability.

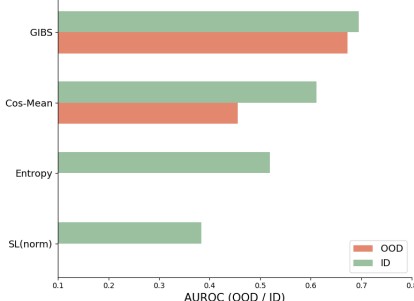

Figure 5: GIBS trained on MoreHopQA and tested on Math without re-training. GIBS consistently outperforms NIBS and white-box baselines under domain shift.

### 5.5.2 ROBUSTNESS TO LABEL SCARCITY (WEAK SUPERVISION).

Our standard setting uses final-answer correctness labels to construct consensus anchors. Here, we ask whether the framework can function without such supervision. We consider two variants: (1) **All Trajectories**, which uses all sampled traces (correct and incorrect) to build consensus and, as shown in Table 4, suffers a clear performance drop due to noise from incorrect paths; and (2) **Self-Consistency**, which replaces gold labels with majority-voted pseudo–correct trajectories. Although still below the oracle *Correct-only* setting, the self-consistency variant recovers much of the per-

formance, indicating that our framework remains effective even when only noisy self-consistency signals are available.

Specifically, on MoreHopQA, the self-consistent trajectories overlap with the gold-correct set by approximately $80\%$, resulting in only a small performance drop relative to *Correct-only*. In contrast, for Llama3.1-8B, the overlap is only about $28\%$, and performance degrades correspondingly. This pattern indicates that when self-consistency produces sufficiently accurate reference trajectories, our method remains effective even without gold labels.

### 5.5.3 ROBUSTNESS TO FREE-FORM REASONING FORMATS

Beyond domain shift and weak supervision, a practical step-wise confidence attribution should also handle *free-form* structured reasoning traces that do not come with explicit graph structures. To assess this aspect, we evaluate our framework on the PRM800K dataset (Lightman et al., 2023c) released by OpenAI, which contains pre-collected GPT-4

|                    | llama3.1-8b | Deepseek | Phi4   |
|--------------------|-------------|----------|--------|
| All trajectories   | 0.5192      | 0.6479   | 0.5546 |
| Self-consistency   | 0.5734      | 0.7843   | 0.6610 |
| Correct-only       | **0.6471**  | **0.8084** | **0.6619** |

Table 4: Comparison of consensus anchor strategies. Correct-only (gold graphs) performs best, while Self-consistency (no gold graphs) provides effective weak supervision with comparable results.

solutions with free-form CoT reasoning and high-quality human step-level annotations.

For each question, we sample $N = 20$ diverse traces and use the gold step labels. NIBS is applied in the same way in the main experiments. For GIBS, each CoT is represented as a linear graph: each sentence-level step is treated as an edge, edges are connected sequentially, and node contents are left empty. We train on 2k traces and evaluate on 10k. Since PRM800K consists of pre-generated GPT-4 outputs, token-level logits are unavailable, and white-box baselines cannot be applied.

Table 5 shows that both NIBS and GIBS achieve strong AUROC, AUCPR, and ACC@80%, demonstrating that our framework transfers effectively to free-form CoT without a special prompt template. NIBS variants perform best overall, while GIBS is competitive but less dominant than in our structured settings, reflecting that PRM800K exposes limited explicit structure for GIBS to exploit.

## 6 CONCLUSION

This paper addresses the important problem of step-wise confidence estimation in LLM reasoning, a key capability for diagnosing where reasoning goes wrong and enabling more reliable model use. We introduced a black-box framework based on the Information Bottleneck principle, with two complementary instantiations: a non-parametric method (NIBS) that assigns confidence via direct consensus alignment, and a graph-based method

| Method           | AUROC ↑ | AUCPR ↑ | ACC@80% ↑ |
|------------------|---------|---------|-----------|
| White-box methods | N/A    | N/A     | N/A       |
| Cos-Max          | 0.6156  | 0.7840  | 0.7734    |
| Cos-Mean         | 0.6821  | 0.8343  | 0.7860    |
| NLI-Max          | 0.7666  | 0.9074  | 0.7899    |
| NLI-Mean         | 0.8181  | 0.9019  | 0.8573    |
| GIBS             | 0.6556  | 0.8203  | 0.7570    |

Table 5: Performance of NIBS and GIBS on the PRM800K dataset with pre-collected GPT-4 traces. White-box methods require token probabilities are not applicable in this setting.

(GIBS) that learns subgraph selection with consensus regularization. Across benchmarks spanning mathematical and textual reasoning, both methods produced accurate and well-calibrated confidence scores, enabling reliable localization of erroneous steps. Moreover, we showed that step-level confidence is not only diagnostic but also actionable: integrating these signals into selective correction improved final-answer accuracy, and GIBS demonstrated promising OOD generalization.

We also acknowledge some limitations of our current work and future directions. First, limited by existing MCS algorithms, the consensus construction is still computationally expensive, limiting scalability to problems with longer and more complex reasoning traces for training. Second, while our framework assigns graded confidence to non-consensus steps, we do not yet explicitly exploit the structure of disagreements, which we leave for future work. Third, our current framework mainly leverages confidence for diagnostics and test-time correction. The step-wise confidence can also be used as important feedback for reinforcement learning, which is underexplored. We believe addressing these challenges will make step-wise confidence attribution more efficient, scalable, and broadly applicable to real-world reasoning tasks.

## 7 ETHICS STATEMENT

We affirm compliance with the ICLR Code of Ethics. This research does not involve human or animal subjects and therefore requires no IRB approval. All datasets are publicly available under appropriate licenses, containing no proprietary or identifiable information, and no attempt was made to re-identify anonymized data. Our methods are developed for academic research and evaluation; applications in safety-critical domains would require additional validation. The work does not disclose harmful insights, and we declare no conflicts of interest or external sponsorship. We acknowledge the presence of possible latent biases in benchmark datasets and recommend caution when generalizing results. No private or sensitive information was accessed, and all experiments comply with institutional and legal requirements. Finally, we maintain research integrity through transparent documentation of methods, hyperparameters, and evaluation protocols, with supplementary materials provided to facilitate reproducibility.

## 8 REPRODUCIBILITY STATEMENT

We have taken deliberate steps to ensure the reproducibility of our results. Section 5.1 of the main text provides a description of the dataset training process, configurations, and evaluation metrics. Additional implementation details and hyperparameter settings are presented in the appendix. All datasets used in this study are publicly available and cited with their respective licenses. To further support replication, we include an anonymized link to the source code and experimental configurations in the supplementary materials. Together, these resources are intended to enable independent researchers to reproduce both the theoretical and empirical findings reported in this work.

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

# A APPENDIX

## A.1 NOTATION TABLE

Table 6: Notation summary used throughout the paper.

| Symbol | Description |
|---|---|
| $x$ | Input problem or question |
| $y_i = (T_i, A_i)$ | $i$-th sampled output: reasoning trajectory $T_i$ and final answer $A_i$ |
| $T_i = \{t_{i1}, t_{i2}, \ldots, t_{iL_i}\}$ | Reasoning trajectory with $L_i$ steps |
| $t_{ij} = (k_{ij}, a_{ij})$ | Step $j$ in trajectory $i$: description $k_{ij}$ and intermediate result $a_{ij}$ |
| $z_i \in \{0, 1\}$ | Correctness label of final answer $A_i$ |
| $c_{ij} \in [0, 1]$ | Confidence score assigned to step $t_{ij}$ |
| $\mathcal{S}$ | Sampled set of reasoning trajectories |
| $G_i = (V_i, E_i)$ | Graph representation of trajectory $T_i$ |
| $v_{ij}, e_{ij}$ | Node (intermediate result) and edge (reasoning operation) for step $t_{ij}$ |
| $G^{MC}$ | Consensus graph constructed from correct trajectories |
| $G^* \subseteq G_i$ | Subgraph selected by GIBS as compressed representation |
| $\mathbf{p}_\theta = \{p_{\theta,ij}\}$ | Soft selection mask predicted by the model |
| $\mathbf{m}_i = \{m_{ij}\}$ | Consensus mask indicating alignment with $G^{MC}$ |
| $X, Z, Y$ | Variables in IB formulation: input trajectory, compressed representation, and correctness |

## A.2 ALGORITHM DETAILS

**NIBS Algorithm.** The procedure of NIBS is shown in Algorithm 1. Given a set of sampled trajectories, we first split them into correct and wrong subsets using the final-answer labels. For a target trajectory, we score each step by comparing it to steps drawn from correct trajectories: a step receives higher confidence if it has strong semantic matches (under the chosen similarity metric) to steps in correct solutions. Within-trajectory similarities are aggregated (e.g., by max or mean) to form the step's score. This produces step-wise confidence without parameter learning and serves as a closed-form IB instantiation that compresses traces to consensus-supported steps.

---

**Algorithm 1:** NIBS

**Input:** Sampled trajectories $\mathcal{S} = \{(T_i, A_i, z_i)\}_{i=1}^N$
**Output:** Step-wise confidence scores $\{c_{ij}\}$
Partition $\mathcal{S}$ into $\mathcal{S}_{\text{correct}} = \{T_i \mid z_i = 1\}$ and $\mathcal{S}_{\text{wrong}}$;
**for** *each trajectory* $T_i = \{t_{i1}, \ldots, t_{iL_i}\}$ **do**
    **for** *each step* $t_{ij}$ *in* $T_i$ **do**
        Compute similarities $\{\text{sim}(t_{ij}, t')\}$ for all steps $t'$ in trajectories from $\mathcal{S}_{\text{correct}}$;
        Aggregate within-trajectory similarities by $\text{Agg}(\cdot)$ (e.g., max or mean);
        Assign $c_{ij}$ by Eq. 2;
    **end**
**end**
**return** $\{c_{ij}\}$

---

**Algorithm 2:** GIBS: Training

**Input:** Reasoning graphs $\{G_i = (V_i, E_i)\}_{i=1}^N$ with labels $\{z_i\}$, model $f_\theta$
**Output:** Trained parameters $\theta$
Partition $\{G_i\}$ into $\mathcal{G}_{\text{correct}} = \{G_i \mid z_i = 1\}$ and $\mathcal{G}_{\text{wrong}}$;
Construct consensus graph $G^{MC}$ from $\mathcal{G}_{\text{correct}}$ (e.g., MCS aggregation);
**for** *each epoch* **do**
    **for** *each* $G_i$ **do**
        Represent each step as $t_{ij} = (v_{ij}, e_{ij})$;
        Align $G_i$ with $G^{MC}$ to obtain consensus mask $\mathbf{m}_i = \{m_{ij}\}$;
        Compute soft mask $\mathbf{p}_\theta = f_\theta(G_i)$ with $p_{\theta,ij} \in [0, 1]$;
        Form soft subgraph $G_i^* = G_i \odot \mathbf{p}_\theta$;
        Compute loss by Equation 5
        Update $\theta$ by gradient descent;
    **end**
**end**
**return** $\theta$

---

**GIBS Algorithm.** The training procedure of NIBS is shown in Algorithm 2. We convert each trajectory into a directed reasoning graph whose steps are represented as edge–node pairs. Using only correct graphs, we construct a consensus graph (e.g., via maximum common subgraph) and align it to each training graph to obtain a consensus mask on steps. In detail, the value for the consensus mask $\mathbf{m}_i$ can be computed as:

$$m_{ij} = \frac{1}{|\mathcal{S}_{\text{correct}}|} \sum_k \mathbf{1}\big[v_{ij} \in \text{MCS}(G_i, G_k)\big],$$

where $\mathbf{1}[\cdot]$ is an indicator function. Thus, $m_{ij} = 1$ if the step consistently appears in all MCS matches with correct graphs, $m_{ij} = 0$ if it never appears, and intermediate values reflect partial support.

A trainable model outputs a soft selection mask over steps; the IB objective is approximated by the sum of a mask-entropy term (encouraging compression) and a cross-entropy term aligning the mask with the consensus anchors (encouraging relevance). We optimize the model parameters by gradient descent to produce calibrated step-selection probabilities.

The inference procedure is shown in Algorithm 3. Given a new trajectory's reasoning graph, the trained model predicts a soft mask over steps. Each step's inclusion probability is reported as its confidence score. Optionally, the soft-masked subgraph can be visualized or passed to downstream routines (e.g., selective correction) to prioritize low-confidence steps while preserving structurally consistent parts of the reasoning.

---

**Algorithm 3:** GIBS: Inference

**Input:** A new reasoning graph $G = (V, E)$, trained model $f_\theta$
**Output:** Step-wise confidence scores $\{c_j\}$
Represent each step as $t_j = (v_j, e_j)$;
Compute soft mask $\mathbf{p}_\theta = f_\theta(G)$;
**for** *each step $t_j$* **do**
  | Set confidence $c_j = p_{\theta,j}$;
**end**
Optionally form $G^* = G \odot \mathbf{p}_\theta$ for visualization or downstream use;
**return** $\{c_j\}$

---

**MCS Algorithm.** MCS algorithm first identifies semantically similar edge pairs between the two reasoning graphs using an NLI model, and selects a small set of high-scoring candidates as seeds. Starting from each seed, it expands the common subgraph iteratively via a BFS, adding only those edges and nodes whose textual entailment scores exceed the thresholds and whose mappings remain consistent. In this paper, we set both thresholds $\tau_v, \tau_e = 0.7$. Among all candidate expansions, the largest resulting subgraph is returned as the MCS along with the induced node and edge mappings. By integrating semantic similarity through entailment scores, the algorithm can align reasoning steps that are lexically different but semantically equivalent.

---

**Algorithm 4:** FindMaximumCommonSubgraph

**Input:** Two reasoning graphs $G_1 = (V_1, E_1), G_2 = (V_2, E_2)$; reasoning texts $\mathcal{R}_1, \mathcal{R}_2$; thresholds $\tau_v, \tau_e$
**Output:** Common subgraph $G^{MC}$, node mapping $\pi_V$, edge mapping $\pi_E$
Generate candidate edge pairs $\mathcal{C} = \{(e_1, e_2) \mid \text{NLI(Entail)}(\mathcal{R}_1(e_1), \mathcal{R}_2(e_2)) \geq \tau_e\}$;
Sort $\mathcal{C}$ by combined edge–node similarity, keep top-$K$ seeds;
**for** *each seed pair $(e_1, e_2) \in \mathcal{C}$* **do**
  | Initialize subgraph $G^*$ with $(e_1, e_2)$, and mappings $\pi_V, \pi_E$;
  | Expand $G^*$ by BFS over neighbors of $e_1$:
  |    For each $e'_1 = (x_1, y_1)$, find best match $e'_2 = (x_2, y_2)$ with entailment $\geq \tau_e$;
  |    If compatible with $\pi_V$, update $G^*$ and mappings;
  | Keep the largest $G^*$ found as the current best;
**end**
Return $G^{MC}$ with similarity annotations, along with $\pi_V, \pi_E$.

---

A.3 COMPLEXITY ANALYSIS OF THE HEURISTIC MCS ALGORITHM

We analyze the time and space complexity of the proposed MCS algorithm. Let $G_1 = (V_1, E_1)$ and $G_2 = (V_2, E_2)$ denote the two input reasoning graphs, and write $m_1 = |E_1|$, $m_2 = |E_2|$, and

$N = m_1 m_2$ for the number of edge pairs. We denote by $\mathcal{T}_{\text{NLI}}$ the time required for a single forward pass of the underlying NLI model, i.e., the cost of computing one entailment-based similarity score between two text segments.

**Time complexity.** The proposed heuristic MCS algorithm proceeds in three phases.

**(1) Pairwise similarity estimation.** In the first phase, the algorithm iterates over the Cartesian product $E_1 \times E_2$ and, for each edge pair $(e_1, e_2)$, evaluates their semantic compatibility. This involves a constant number of calls to the NLI model to score the correspondence between the edges and their incident nodes. Thus the total cost of this phase is

$$O\big(N \cdot \mathcal{T}_{\text{NLI}}\big) \;=\; O\big(|E_1|\,|E_2| \cdot \mathcal{T}_{\text{NLI}}\big).$$

**(2) Candidate ranking and pruning.** Among all edge pairs that satisfy the similarity thresholds, the algorithm ranks them according to their combined edge–node similarity and retains only the top-$K$ pairs (with $K = 10$ in our experiments) as seed candidates. If we denote by $M \leq N$ the number of pairs that pass the thresholds, then sorting these candidates by score requires

$$O\big(M \log M\big) \;\subseteq\; O\big(N \log N\big) \;=\; O\big(|E_1|\,|E_2| \log(|E_1|\,|E_2|)\big)$$

time in the worst case. After sorting, truncating the list to the top-$K$ pairs is $O(1)$.

**(3) Heuristic subgraph expansion.** Starting from each of the at most $K$ seed pairs, the algorithm performs a BFS-style expansion on $G_1$, greedily adding neighboring edges and nodes as long as they admit compatible matches in $G_2$ within the pre-filtered candidate set. For a fixed seed, this expansion touches at most $O(|V_1| + |E_1|)$ graph elements of $G_1$, and since $K$ is a constant independent of the input sizes, the total cost of this phase is

$$O\big(K(|V_1| + |E_1|)\big) \;=\; O\big(|V_1| + |E_1|\big).$$

Combining these contributions, the overall running time of the heuristic MCS algorithm can be bounded as

$$\mathcal{T}_{\text{total}} \;=\; O\big(|E_1|\,|E_2| \cdot \mathcal{T}_{\text{NLI}} + |E_1|\,|E_2| \log(|E_1|\,|E_2|) + |V_1| + |E_1|\big).$$

In practice, the pairwise similarity phase is dominant, because it invokes computationally expensive NLI forward passes for $O(|E_1|\,|E_2|)$ edge pairs. Treating the NLI architecture and the hyperparameter $K$ as fixed, the overall complexity with respect to the graph sizes is thus essentially quadratic in the number of edges:

$$\mathcal{T}_{\text{total}} = O\big(|E_1|\,|E_2| \cdot \mathcal{T}_{\text{NLI}}\big).$$

**Space complexity.** The additional memory footprint is dominated by storing candidate edge pairs and the intermediate MCS structures. In the worst case, when many edge pairs pass the similarity thresholds, the algorithm keeps $M = O(|E_1|\,|E_2|)$ candidates together with their scores. The data structures used for BFS-based expansion are linear in the size of $G_1$, i.e., $O(|V_1| + |E_1|)$, and are reused across different seeds. Therefore, the overall auxiliary space complexity is

$$\mathcal{O}_{\text{space}} \;=\; O\big(M + |V_1| + |E_1|\big) \;\subseteq\; O\big(|E_1|\,|E_2|\big).$$

The parameters and internal buffers of the NLI model contribute only a constant term with respect to the graph sizes, and do not affect the asymptotic behavior.

## A.4 EXPERIMENT DETAILS

**Datasets** We evaluate our methods on three datasets spanning both mathematical and textual reasoning tasks: (1) **GSM8K** Cobbe et al. (2021): A benchmark of grade-school math word problems, widely used for evaluating numerical reasoning and basic arithmetic logic. (2) **Math** Hendrycks et al. (2021): A more challenging dataset of competition-level mathematics problems. This dataset stresses the scalability of our methods to more complex mathematical reasoning. (3) **More-HopQA** Schnitzler et al. (2024): A multi-hop question answering dataset, where solving a query requires integrating evidence across multiple passages. MoreHopQA focuses on textual compositional reasoning and tests whether our framework generalizes to non-mathematical domains.

### A.4.1 BASELINE

We compare against four white-box CE baselines, each adapted to the step-wise reasoning setting:

●White-box methods: (1) Normalized Sequence Likelihood (**SL(norm)**) Lin et al. (2024); Cole et al. (2023). For each reasoning step $t_{ij}$, we concatenate the description $k_{ij}$ and the intermediate result $a_{ij}$ together. The step-wise confidence is then defined as the SL(norm) of all tokens in this step:$c_i^{SL} = \frac{1}{|T_i|} \sum_{t \in T_i} \log p_\theta(x_t|x_{<t})$. (2) Token-level Entropy (**Entropy**) Kuhn et al. (2023): Token Entropy computes the entropy of the predictive distribution at each token and then averages over tokens to obtain step-wise uncertainty, which is defined as $H_{\text{step}} = \frac{1}{|T|} \sum_{t=1}^{T} \sum_{k=1}^{V} p_{t,k} \log p_{t,k}$. Confidence is then taken as the $-H_{step}$. (3) **P(true)** Kadavath et al. (2022): Following prior work, we directly query the LLM itself to judge whether the current step is correct or not, and then use the probability assigned to the label *true* as the confidence score. The prompt we used is in Appendix A.4.4. (4) **LeCo** Yao et al. (2024): We also include LeCo, which introduces a logit-based confidence scoring method. It combines three components: the average token score, the step divergence score, and the inter-step transition score to compute an overall step confidence.

● NIBS: These baselines implement equation 2, where a similarity function is applied between steps and then aggregated into a confidence score. We use two similarity functions: cosine similarity and natural language inference (NLI), combined with three aggregation strategies: mean, max, and random. Cos-Mean, Cos-Max, NLI-Mean, NLI-Max.

●GIBS: Our proposed model that learns a compressed subgraph representation aligned with consensus reasoning structures.

### A.4.2 CONFIGURATIONS

For all experiments, we evaluate three representative LLMs: LLaMA-3.1-8B-Instruct, Phi-4-Reasoning, and DeepSeek-R1-Distill-Qwen-32B, and use identical prompting templates (Appendix A.4.4) to ensure fair comparisons. Decoding is performed with temperature set to 1.0, and for each input question, we sample $N = 20$ reasoning traces.

To compute semantic similarity in our non-parametric baselines, NIBS uses cosine similarity over sentence embeddings from `bert-base-uncased` (Devlin et al., 2019), and both NIBS and GIBS use NLI-based similarity scores from the `DeBERTa-large-MNLI` model (He et al., 2020). For GIBS, we extract reasoning graphs via *structured generation*: a LangFun-style template (Appendix A.4.4) asks the model to instantiate a Python `ReasoningGraph` class, and we recover nodes and edges with a simple rule-based parser. The edge- and node-level entailment thresholds $\tau_e, \tau_v$ used in the MCS procedure are both fixed to 0.7, following prior work, and a sensitivity analysis is provided in Appendix A.4.5.

For the learned method, we train the GIBS model on 2k reasoning graphs constructed from a mixture of correct and incorrect trajectories. The model uses a 2-layer GCN encoder with hidden dimension 128 and dropout 0.1, and is optimized with Adam (learning rate $10^{-3}$) with early stopping based on validation loss. Gold step-level correctness labels for computing AUROC, AUCPR, and ACC@C% are obtained from GPT-4o using the evaluation templates in Appendix A.4.4. All reported results are averaged over 10k reasoning traces per dataset. Experiments are implemented in PyTorch and run on a single NVIDIA A100 GPU.

### A.4.3 EVALUATION METRICS

We adopt three complementary metrics to evaluate the effectiveness of our approach: AUROC, AUCPR, and ECE. (1) **AUROC** (Area Under the Receiver Operating Characteristic Curve) Lin et al. (2023) measures the model's ability to discriminate between correct and wrong steps across all thresholds, reflecting overall ranking performance. (2) **AUCPR** (Area Under the Precision–Recall Curve) Davis & Goadrich (2006) focuses on ranking under class imbalance, which is crucial since erroneous steps typically form a small fraction of all steps. (3)**ACC@c%** Geifman & El-Yaniv (2017) is a standard metric in selective prediction, which reports the accuracy when retaining only the top-c% most confident predictions. A reliable CE method will get a higher ACC@c% by ranking correct predictions above incorrect ones.

(4) **ECE** (Expected Calibration Error) assesses the calibration of step-level confidence scores by comparing predicted confidence with empirical accuracy across bins, with lower values indicating better alignment.

### A.4.4 PROMPT TEMPLATE & FEW SHOT EXAMPLES

---

**Prompt Template for Structured LLM Responses**

**System Prompt:** Your role as an assistant involves thoroughly exploring questions through a systematic thinking process before providing the final precise and accurate solutions.

**Prompt:**
```
Please respond to the last INPUT_OBJECT with OUTPUT_OBJECT according to OUTPUT_TYPE.

INSTRUCTIONS:
- Do NOT define or repeat any class or function.
- ONLY produce an OUTPUT_OBJECT that instantiates the OUTPUT_TYPE.
- The output must be valid Python using the given type names.
- Do NOT generate code, explanation, or helper variables.
- Only output an object like ReasoningGraph(...).

INPUT_OBJECT:
1 + 1 =

OUTPUT_TYPE:
Answer
```

```python
class Answer:
    final_answer: int
```

```
OUTPUT_OBJECT:
```
```python
Answer(
final_answer=2
)
```

```
INPUT_OBJECT:
question

OUTPUT_TYPE:
ReasoningGraph
```

```python
class ReasoningNode:
    id: int
    description: str
    output: Union[int, float, str]
    depends_on: list[int]

class ReasoningGraph:
    nodes: list[ReasoningNode]
    final_answer: Union[int, float, str]
```

```
OUTPUT_OBJECT:
"""
```

---

---

**Regeneration Template Final Answer Feedback**

**Structured LLM Responses Prompt + Prompt:**
# NOTE: The previous final answer '{answer}' is incorrect.

Please regenerate the OUTPUT_OBJECT only.

Do not provide any reasoning, explanation, or extra text.

The OUTPUT_TYPE remains ReasoningGraph.

---

**Regeneration Template Step-wise Feedback**

**Structured LLM Responses Prompt + Prompt:**
# FEEDBACK:

The previous final answer '{previous_answer}' is incorrect.

## Reasoning Process:

{reasoning_process}

## Identified Errors (steps with low confidence):

{error_description}

Do not follow the identified error steps. Please reanalyze and regenerate a correct final answer that is different from the previous incorrect answer which is {previous_answer}. Return only the Reasoning-Graph object without any explanations.

---

**GPT Evaluation Prompt**

**System Prompt:** You are evaluating a mathematical reasoning graph for correctness. Given the problem, correct answer, and a series of reasoning steps (edge-node pairs), determine if each step is mathematically and logically correct.

**Prompt:**
Math Problem: {question}
Correct Answer: {answer}
Reasoning Graph (Edge-Node Pairs):
{pairs_text}
Instructions:
1. Evaluate each edge-node pair in the context of solving this problem
2. Consider if the edge description accurately describes what is being calculated
3. Check if the node value is mathematically correct given the edge description
4. Verify that each step logically follows from the problem or previous steps
For each pair, respond with:
- 1 if the edge-node pair is correct
- 0 if the edge-node pair is incorrect
Format your response as a comma-separated list of digits (no spaces), one digit per pair, without any explanation.
Example for 5 pairs: 1,0,1,1,0
Your evaluation:

---

## A.4.5 SENSITIVITY STUDY

We investigate the robustness of our framework with respect to two key hyperparameters: (i) the NLI-based similarity thresholds used in MCS construction, and (ii) the number of sampled trajectories $N$ used to form the consensus.

| Threshold | 0.5 | 0.6 | 0.7 | 0.8 | 0.9 |
|---|---|---|---|---|---|
| AUROC | 0.5821 | 0.6081 | 0.6619 | 0.6658 | 0.6112 |

Table 7: Sensitivity of AUROC to NLI thresholds on MoreHopQA with Phi-4.

| # Samples $N$ | 5 | 10 | 15 | 20 | 25 |
|---|---|---|---|---|---|
| AUROC | 0.5964 | 0.6024 | 0.6569 | 0.6619 | 0.6679 |

Table 8: Sensitivity of AUROC to the number of sampled trajectories $N$ on MoreHopQA with Phi-4.

**Sensitivity of NLI thresholds.**    In all main experiments, the entailment thresholds $\tau_e$ and $\tau_v$ are set to $0.7$, following prior work on semantic alignment (Lin et al., 2023; Da et al., 2025). To assess sensitivity to this choice, we conduct a sensitivity analysis on MoreHopQA with Phi-4, jointly varying both thresholds in the range $[0.5, 0.9]$. As shown in Table 7, the AUROC remains stable when $\tau_e, \tau_v$ lie in $[0.7, 0.8]$, and we observe noticeable degradation only when the thresholds are set too strictly ($> 0.8$) or too loosely ($< 0.7$). This supports our default choice $\tau_e, \tau_v = 0.7$ as a robust operating point.

**Sensitivity of the number of sampled trajectories.**    We also evaluate how performance depends on the number and diversity of sampled trajectories $N$ used for consensus construction. Table 8 reports AUROC on MoreHopQA with Phi-4 as we vary $N$ from 5 to 25. The results show that performance improves as $N$ increases and stabilizes once sufficient diversity is reached (approximately $N > 15$), indicating that the consensus becomes more reliable with richer sampling but remains robust beyond this point.

In principle, black-box confidence estimation methods approximate the underlying output distribution by sampling multiple trajectories; hence, performance naturally scales with $N$ before plateauing, a behavior consistently observed in prior work (Kuhn et al., 2023; Lin et al., 2023). Following these works and our empirical analysis, we set $N = 20$ in all main experiments as a good trade-off between computational efficiency and stability.

**Sensitivity of GNN backbones.**    Our learned method GIBS uses a graph neural network as a backbone to encode consensus-aware features. In the main experiments we adopt a standard 2-layer GCN. To assess the role of the backbone architecture, we also experimented with GraphSAGE, GAT, and GIN on MoreHopQA with Phi-4 under the same training protocol (Table 9). We find that all backbones yield comparable behavior in this setting, and a simple GCN already provides a strong and reliable choice for modeling the required structural patterns.

## A.5    CASE STUDY

Figure 6 provides a case study from the MoreHopQA dataset that qualitatively demonstrates the advantage of our method. Initially, the LLM produces an incorrect answer by focusing on the wrong entity (the magazine's founding year instead of the publisher's). Our method correctly identifies this flawed premise by assigning low confidence to the initial reasoning steps. While simple final-answer feedback is insufficient for the model to find this root error, our step-wise feedback explicitly flags the low-confidence steps. This targeted guidance prompts the model to reconsider its flawed premise, revise its reasoning trajectory, and arrive at the correct answer. This case illustrates that LLMs can effectively self-correct when their reasoning uncertainty is accurately localized.

## A.6    FIRST-ERROR STEP DETECTION

A natural concern is that step-wise confidence attribution might be dominated by *later* erroneous steps: once a trajectory has already gone off course, subsequent steps often become trivially inconsistent with the correct reasoning pattern. Thus, for debugging and test-time correction, identifying the *first* wrong step is most critical, since all downstream errors are typically propagated from this point.

| GNN Backbone | GCN | GraphSAGE | GAT | GIN |
|---|---|---|---|---|
| AUROC | 0.6619 | 0.6446 | 0.5806 | 0.5815 |

Table 9: Influence of different GNN backbones on AUROC (MoreHopQA with Phi-4).

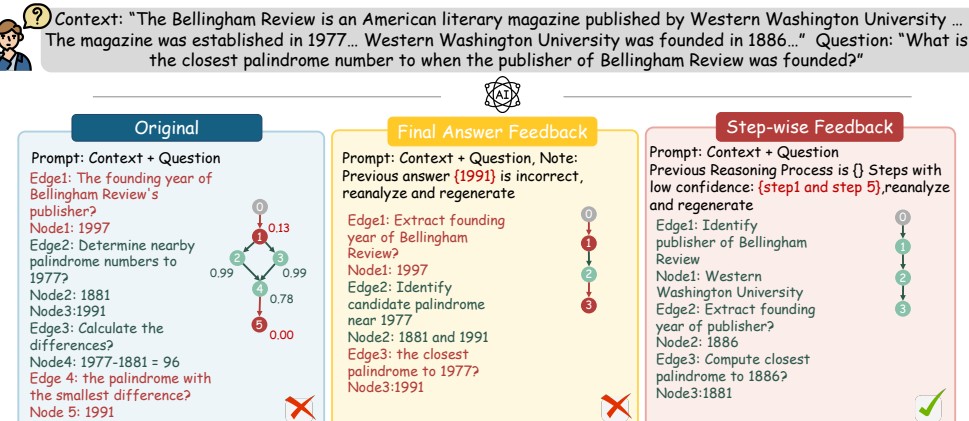

Figure 6: A case study on the MoreHopQA dataset comparing the effect of different feedback types. Providing targeted, step-wise feedback on low-confidence reasoning steps is effective at guiding the model to correct its root error, whereas providing simple final-answer feedback is not.

Our framework inherently assigns confidence scores to *all* intermediate steps, so it also produces a score for the earliest erroneous step in each trace. To disentangle performance on the true failure point from that on subsequent steps, we conduct an additional evaluation that focuses exclusively on the first error. Concretely, for each trajectory that contains at least one incorrect step, we locate the first step whose correctness label is 0 and keep *only* this step when computing the metrics; all later steps in that trajectory are excluded from the evaluation set. We then recompute AUROC and an *Adaptive F1* metric, where Adaptive F1 is defined as the maximum F1 score obtained by sweeping a threshold over the predicted confidence scores.

Table 10 summarizes the results on MoreHopQA for three LLMs. Across all models, GIBS achieves the highest AUROC, indicating that it is particularly effective at ranking the first erroneous step above the correct ones. Its Adaptive F1 is also competitive, and in the case of DeepSeek-R1, it attains the best F1 among all methods. Compared to token-level baselines such as P(true), SL(norm), and Entropy, as well as our non-parametric NIBS variants (Cos-Max/Mean, NLI-Max/Mean), GIBS exhibits a clear advantage in detecting the initial divergence point.

| | Llama3.1-8b | | Deepseek | | Phi4 | |
|---|---|---|---|---|---|---|
| | AUROC | Adaptive F1 | Auroc | Adaptive F1 | AUROC | Adaptive F1 |
| P(true) | 0.5126 | **0.8400** | 0.5152 | 0.8518 | 0.5090 | 0.8750 |
| SL(norm) | 0.4941 | 0.8376 | 0.3009 | 0.8518 | 0.4724 | 0.8749 |
| Entropy | 0.5570 | 0.8376 | 0.6004 | 0.8518 | 0.6837 | 0.8748 |
| LECO | 0.3839 | 0.8363 | 0.3307 | 0.8332 | 0.3884 | 0.8723 |
| Cos-Max | 0.4275 | 0.8366 | 0.4529 | 0.8655 | 0.4772 | 0.8871 |
| Cos-Mean | 0.4863 | 0.8366 | 0.5313 | 0.8655 | 0.5099 | 0.8871 |
| NLI-Max | 0.5375 | 0.8366 | 0.7110 | 0.8655 | 0.6632 | 0.8883 |
| NLI-Mean | 0.5285 | 0.8366 | 0.6544 | 0.8655 | 0.5824 | 0.8868 |
| GIBS | **0.6164** | 0.8376 | **0.7885** | **0.8681** | **0.6841** | **0.8786** |

Table 10: First-error step detection performance (AUROC and Adaptive F1) on MoreHopQA. Adaptive F1 is the best F1 over thresholds on the confidence scores.

## B  LLM Usage Statement

In accordance with the ICLR 2026 submission guidelines, we disclose that large language models (LLMs) were used only for language editing and proofreading of this draft.

