# OpenReview forum: "Where Reasoning Fails: Step-wise Confidence Attribution in Black-box LLMs"
_ICLR.cc/2026/Conference — Submitted to ICLR 2026_

### Official Review · Reviewer_2s99 · 2025-10-28

**Soundness:** 2
**Presentation:** 2
**Contribution:** 2
**Rating:** 2
**Confidence:** 4

**Summary:**

This work proposes methods on step-wise confidence estimation of black-box LLMs in reasoning tasks. Two confidence estimation are proposed, which are based on measuring the similarity of different steps among various sampled reasoning traces. Empirical evaluations are conducted against various baseline methods, where the proposed methods show better performance. Moreover, the utility of the proposed methods is demonstrated in guiding LLM self-correction, which is more effective than self-correction based on final prediction correctness only.

**Strengths:**

1. The proposed step-wise confidence estimation methods are principled and intuitive, which could be effective given strong implementations.

2. The empirical evaluation is thorough, which aims to demonstrate the effectiveness of the proposed methods through direct evaluations and downstream applications.

**Weaknesses:**

1. In general, this work lacks clarity in its problem formulation, descriptions of evaluation setup and implementation details. Specific issues are mentioned in the next sections.

2. The introduced confidence estimation methods introduce dependency on external models and/or additional requirements on the format of model outputs, which could make them brittle and hard-to-generalize.

    (a) The NIBS method requires either BERT embeddings or an NLI model. Therefore, its effectiveness depends on the reliability of the embeddings and/or the NLI model. (There is no clear description regarding the BERT embeddings and the NLI model. Which BERT was used? Was the NLI model trained? Which model is the NLI model based on?)

   (b) The GIBS method requires the model to output structured reasoning traces or a post-processor for parsing the reasoning trace. (There is no specific information in the paper on which approach was actually used.) If the former approach is used, it requires the reasoning model to be capable enough to generate structured reasoning traces and might alter model behavior and introduce computational overhead. If the latter is used, an additional post-processor is required for the method. Moreover, the GIBS method also requires an NLI model. (Again, the specific information of the used NLI model is lacking). For annotating the entailment, two hyper-parameters are introduced ($\tau_{e}$, $\tau_{v}$), of which the values seem to be arbitrarily set (Line 787). How sensitive is the method regarding the values of these?

3. The evaluation setup and the main objective of the step-wise confidence estimation should be better clarified.

    (a) Is there a definition of "gold-standard" step-wise confidence? Is it the LLM-predicted probability of the steps? This is critical since it determines how the confidence estimation methods should be evaluated.

    (b) Related to (a), in Section 5.2 (Table 1), how are the AUROC and ECE, etc. computed? These metrics require accuracy. Is the accuracy computed at the step level? If it is, how is the step-level accuracy computed?

**Questions:**

Please see the Weaknesses section.

---

> ### Author Response · Authors · 2025-11-28
> **Response to Reviewer 2s99: W1(a) and (b)**
>
> > W1(a) The NIBS method requires either BERT embeddings or an NLI model. Therefore, its effectiveness depends on the reliability of the embeddings and/or the NLI model. (There is no clear description regarding the BERT embeddings and the NLI model. Which BERT was used? Was the NLI model trained? Which model is the NLI model based on?) Moreover, the GIBS method also requires an NLI model.
>
> - **[Implementation Details]** We thank the reviewer for pointing out the missing implementation details. For BERT-based similarity, we use cosine similarity computed from the **bert-base-uncased model** [1]. For NLI-based similarity, we use the **publicly available DeBERTa-large-MNLI model** [2]. Both models are off-the-shelf and not additionally trained. We have included these details **in Section 5.1 and A.4.2**.
>
> - **[Semantic Evaluation]**: We clarify that in black-box settings, where token logits are unavailable, using external models for semantic consistency evaluation is **both standard and necessary for consistency checking**. Because LLM outputs can convey the same meaning through **diverse surface forms**, a semantic-level comparison is required and cannot rely on exact string matching. Prior work has consistently used **embedding cosine similarity [3,4,5,6] and NLI models [7,8]** to assess semantic equivalence. We follow this practice and use off-the-shelf models without additional training, ensuring the pipeline is **reproducible and not computationally prohibitive**.
>
> > W1(b)The GIBS method requires the model to output structured reasoning traces or a post-processor for parsing the reasoning trace. (There is no specific information in the paper on which approach was actually used.) If the former approach is used, it requires the reasoning model to be capable enough to generate structured reasoning traces and might alter model behavior and introduce computational overhead. If the latter is used, an additional post-processor is required for the method.
>
> - **[Clarification]**: We thank the reviewer for raising this point. **For GIBS, we use structured generation via prompting templates rather than a post-processing parser**. Following prior work [9,10,11], we adopt the **LangFun prompt template** [12] (Appendix A.3.4) to elicit a Pydantic-style ReasoningGraph Python class that encodes step-level reasoning in a JSON-like format. The structure is then extracted with **simple rule-based parsing**. This procedure is lightweight and does not affect model behavior, as it only adds minimal formatting instructions (similar to JSON-style CoT prompting). We have clarified these implementation details **in Section 4.3 and 5.1** of the revised version.
>
> > For annotating the entailment, two hyper-parameters are introduced ($\tau_e$, $\tau_v$), of which the values seem to be arbitrarily set (Line 787). How sensitive is the method regarding the values of these?
>
> - **[Prior Work]:** The threshold values $\tau_e$ and $\tau_v$ were set to 0.7, following prior work[8, 14] on semantic alignment.
> - **[New Sensitivity Analysis]:** To fully address your concern, we conducted a **sensitivity analysis** on the **MoreHopQA dataset with Phi4**, varying both thresholds in the range **[0.5, 0.9]**. As shown in Table 2, the performance remained **stable within the $[0.7, 0.8]$ range**. We observed noticeable degradation only when the thresholds were set too strictly (>0.8) or too loosely (<0.7). We have included these sensitivity study results **in Appendix A.4.5**.
>
> **Table 2.** Performance across NLI thresholds on the MoreHopQA dataset with Phi4.
>
> | threshold | 0.5 | 0.6 | 0.7 | 0.8 | 0.9 |
> | --- | --- | --- | --- | --- | --- |
> | AUROC | 0.5821 | 0.6081 | 0.6619 | 0.6658 | 0.6112
>  |

---

> ### Author Response · Authors · 2025-11-28
> **Response to Reviewer 2s99: W1(a) and (b) citation**
>
> [1] Devlin, Jacob, et al. "Bert: Pre-training of deep bidirectional transformers for language understanding." *Proceedings of the 2019 conference of the North American chapter of the association for computational linguistics: human language technologies, volume 1 (long and short papers)*. 2019.
>
> [2] He, Pengcheng, et al. "DEBERTA: DECODING-ENHANCED BERT WITH DISENTANGLED ATTENTION." *International Conference on Learning Representations*.
>
> [3]Wang, Yuxia, et al. "Uncertainty Estimation and Reduction of Pre-trained Models for Text Regression." *Transactions of the Association for Computational Linguistics* 10 (2022): 680-696.
>
> [4]Reimers, Nils, and Iryna Gurevych. "Sentence-BERT: Sentence Embeddings using Siamese BERT-Networks." *Proceedings of the 2019 Conference on Empirical Methods in Natural Language Processing and the 9th International Joint Conference on Natural Language Processing (EMNLP-IJCNLP)*. 2019.
>
> [5] Nguyen, Dang, Ali Payani, and Baharan Mirzasoleiman. "Beyond Semantic Entropy: Boosting LLM Uncertainty Quantification with Pairwise Semantic Similarity." *arXiv preprint arXiv:2506.00245* (2025).
>
> [6]Ye, Zihuiwen, et al. "Uncertainty-Aware Step-wise Verification with Generative Reward Models." *ICLR Workshop: Quantify Uncertainty and Hallucination in Foundation Models: The Next Frontier in Reliable AI*.
>
> [7] Kuhn, Lorenz, Yarin Gal, and Sebastian Farquhar. "Semantic Uncertainty: Linguistic Invariances for Uncertainty Estimation in Natural Language Generation." *The Eleventh International Conference on Learning Representations, 2023*
>
> [8] Lin, Zhen, Shubhendu Trivedi, and Jimeng Sun. "Generating with Confidence: Uncertainty Quantification for Black-box Large Language Models." *Transactions on Machine Learning Research*.
>
> [9]Lin, Chu-Cheng, et al. "Type-Compliant Adaptation Cascades: Adapting Programmatic LM Workflows to Data." *arXiv preprint arXiv:2508.18244* (2025).
>
> [10]Yang, Ziran, et al. "Understanding the Sources of Uncertainty for Large Language and Multimodal Models." *ICLR Workshop: Quantify Uncertainty and Hallucination in Foundation Models: The Next Frontier in Reliable AI*.
>
> [11]Liu, Jiarun, et al. "JoyAgent-JDGenie: Technical Report on the GAIA." *arXiv preprint arXiv:2510.00510* (2025).
>
> [12]https://github.com/google/langfun
>
> [13] Lightman, Hunter, et al. "Let's verify step by step." *The Twelfth International Conference on Learning Representations*. 2023.
>
> [14] Da, Longchao, et al. “*Understanding the Uncertainty of LLM Explanations: A Perspective Based on Reasoning Topology*.” *Second Conference on Language Modeling*, 2025.

---

> ### Author Response · Authors · 2025-11-28
> **Response to Reviewer 2s99: W2(a) and (b)**
>
> > W2(a) The evaluation setup and the main objective of the step-wise confidence estimation should be better clarified. (a) Is there a definition of "gold-standard" step-wise confidence? Is it the LLM-predicted probability of the steps? This is critical since it determines how the confidence estimation methods should be evaluated.
>
> - **[Clarification]** We clarify that while **no gold-standard step-wise confidence (e.g., token-level probabilities) exists**, we do have step-level correctness labels as described in Section 4.4 and Appendix A.4.4. We instead use final-answer correctness as semi-supervision and derive consensus anchors from correct reasoning trajectories to approximate the latent “correctness” structure. Step-level confidence is therefore an **attribution-based reliability measure indicating alignment with these anchors**, rather than an LLM-predicted probability. For evaluation, we **treat steps overlapping with the consensus as positive (reliable) and others as negative**, and report AUROC/AUPRC/ECE to assess how well each method separates structurally consistent (reliable) from inconsistent (error-prone) steps.
>
> - **[LLM-Predicted Step Probability]** We note that LLM-predicted step probabilities are included in our baselines **(e.g., SL(norm), Entropy; see Appendix A.4.1)** as predicted confidence. All methods are evaluated under the same protocol, measuring how well their predicted scores distinguish reliable (consensus-aligned) from unreliable (inconsistent) steps
>
> > W2(b) Related to (a), in Section 5.2 (Table 1), how are the AUROC and ECE, etc. computed? These metrics require accuracy. Is the accuracy computed at the step level? If it is, how is the step-level accuracy computed?
>
> - **[Step-Level Ground Truth]** We thank the reviewer for the question. To compute evaluation metrics (e.g., AUROC, ACC@80%), we require step-level correctness labels. As described in Section 4.4 and Appendix A.3.4, we follow prior work [1,2,3,4] and use an LLM-as-a-Judge approach (e.g., GPT-4o) to assess each intermediate step against the gold final answer. These LLM-judged labels serve as a proxy ground truth for step-level accuracy. AUROC and ECE will evaluate how well the predicted confidence scores align with these correctness labels.
>
> We hope these clarifications and proposed revisions address the reviewer's concerns.
>
> [1]Ye, Zihuiwen, et al. "Uncertainty-Aware Step-wise Verification with Generative Reward Models." *ICLR Workshop: Quantify Uncertainty and Hallucination in Foundation Models: The Next Frontier in Reliable AI*.
>
> [2] Lin, Zhen, Shubhendu Trivedi, and Jimeng Sun. "Generating with Confidence: Uncertainty Quantification for Black-box Large Language Models." *Transactions on Machine Learning Research*.
>
> [3]Xiong, Miao, et al. "Can LLMs Express Their Uncertainty? An Empirical Evaluation of Confidence Elicitation in LLMs." *The Twelfth International Conference on Learning Representations*.
>
> [4]Huang, Zhiqi, et al. "Confidence-Based Response Abstinence: Improving LLM Trustworthiness via Activation-Based Uncertainty Estimation." *Proceedings of the 2nd Workshop on Uncertainty-Aware NLP (UncertaiNLP 2025)*. 2025.

---

### Official Review · Reviewer_Jq1V · 2025-10-31

**Soundness:** 3
**Presentation:** 3
**Contribution:** 3
**Rating:** 6
**Confidence:** 3

**Summary:**

- This paper proposes two confidence attribution methods called NIBS and GIBS. NIBS is a non-parametric method while GIBS employs graph-based modeling method to check the confidence of the steps in the reasoning chain. They are based on the information bottleneck principle. Authors conduct experiments on several LLM backbones and different reasoning datasets to present the effectiveness of their design.

**Strengths:**

- This is a novel paper discuss an interesting problem Confidence Attribution, which is important for LLM reasoning. Authors design new black-box methods to address the disadvantages of current white-box methods.
- The design of NIBS and GIBS demonstrates strong innovation, and the authors have conducted a substantial amount of work in their experiments.

**Weaknesses:**

- Regarding the process of constructing a graph, the author provides a rather brief introduction in the text. I believe this process could be explained more effectively with the inclusion of additional concrete examples.
- In my understanding, the model designed by the authors should be a GNN. Based on this, they developed subsequent training and inference methods for NIBS. However, the paper does not elaborate on specific details such as the choice of GNN backbone. I only confirmed this after reviewing the code.
- The method is tested under several datasets which all have definite answer labels, which may be unrealistic in certain scenarios (e.g., open-domain generation). Although the authors note this is a reasonable assumption for diagnostic tasks, they do not discuss the impact of label noise or partially correct answers.

**Questions:**

- Can the author explore how different graph construction methods impact performance? For instance, what performance differences arise when processing reasoning steps as homogeneous graphs vs heterogeneous graphs?
- Did the authors test the impact of different GNN backbones on performance? Based on the code, it appears they only experimented with the GCN model.
- Can the research topic being studied by the author positively impact RL-based LLM post-training? For example, by utilizing the model you propose to estimate the confidence level of the reasoning chain during the RL rollout process, thereby yielding additional benefits.

---

> ### Author Response · Authors · 2025-11-28
> **Response to Reviewer Jq1V: W1,W2 and W3**
>
> We sincerely thank the reviewer Jq1V for your time and constructive feedback. We are glad that you found confidence attribution for LLM reasoning to be an important and novel problem, and that you viewed our proposed black-box approaches (NIBS and GIBS) and the accompanying extensive experiments as innovative and thorough. Below we provide clarifications on your concerns.
>
> > W1: Regarding the process of constructing a graph, the author provides a rather brief introduction in the text. I believe this process could be explained more effectively with the inclusion of additional concrete examples.
>
> - Thank you for pointing this out. We clarify that **Figure. 1** provides a concrete example of how we construct the reasoning graph. In practice, we elicit the structured set of sub-questions and answers via a LangFun-style prompting setup (the full prompt is given in Appendix A.4.4), and then extract the reasoning graph using a simple rule-based parser. In the three reasoning graphs shown in Figure 1, dark-colored sub-questions correspond to edge sentences, and light-colored answers correspond to nodes; the structural information (i.e., which nodes are connected) is determined by the dependency relations in the formatted outputs from LLMs. We have revised **Section. 4.4** to describe this procedure and explicitly reference Figure. 1.
>
> > W2: In my understanding, the model designed by the authors should be a GNN. Based on this, they developed subsequent training and inference methods for NIBS. However, the paper does not elaborate on specific details such as the choice of GNN backbone. I only confirmed this after reviewing the code.
>
> - **[GIBS Architecture Clarification]:** We clarify that the GNN architecture is used **only in GIBS**; NIBS is a non-parametric method based solely on semantic similarity and does not involve any neural network training. For GIBS, consensus anchors are constructed by **extracting embeddings from the maximum consensus subgraph**, for which we use a **standard 2-layer GCN backbone**. We updated **Appendix A.4.2** to include these architectural details to ensure reproducibility without relying on the codebase. Thanks for you pointing out this concern.
>
> > W3: The method is tested under several datasets, which all have definite answer labels, which may be unrealistic in certain scenarios (e.g., open-domain generation). Although the authors note this is a reasonable assumption for diagnostic tasks, they do not discuss the impact of label noise or partially correct answers.
>
> - **[Existing MorehopQA Dataset]** We would like to clarify that not all datasets used in our experiments have definite answer labels. In particular, MoreHopQA is an open-ended, multi-hop QA dataset where the model generates free-form answers from multiple context paragraphs. For this dataset, we follow standard evaluation protocols [1,2] and use an LLM-as-a-judge to annotate correctness, which naturally introduces label noise. Despite this, our methods remain robust and effective, suggesting applicability beyond settings with deterministic ground-truth answers.
>
> - **[New Experiments]** Our framework can also operate **without explicit gold answers**: the IB objective itself does not require supervised correctness signals, and in weakly supervised settings we can replace gold labels with self-consistency–based agreement among sampled trajectories, using **majority-voted trajectories to construct consensus anchors**. We instantiate this variant by training with self-consistency instead of gold labels. As shown in Table 1, on MoreHopQA the self-consistent trajectories overlap with gold-correct traces by ≈80%, leading to only a modest AUROC drop. When the overlap is lower (≈28% for Llama3-8B), performance degrades more, which matches the intuition that the quality of self-consistency pseudo-labels is critical.
>
> **Table 1:** Performance under semi-supervision vs. self-consistency
>
>  |  | Llama3-8b | Deepseek | Phi4 |
> | --- | --- | --- | --- |
>  | Correct-only | 0.6471 | 0.8084 | 0.6619 |
>   | Self-consistency | 0.5734 | 0.7843 | 0.6610 |
>
> [1]Ye, Zihuiwen, et al. "Uncertainty-Aware Step-wise Verification with Generative Reward Models." *ICLR Workshop: Quantify Uncertainty and Hallucination in Foundation Models: The Next Frontier in Reliable AI*.
>
> [2]Lin, Zhen, Shubhendu Trivedi, and Jimeng Sun. "Generating with Confidence: Uncertainty Quantification for Black-box Large Language Models." *Transactions on Machine Learning Research*.

---

> ### Author Response · Authors · 2025-11-28
> **Response to Reviewer Jq1V: Q1,Q2 and Q3**
>
> > Q1: Can the author explore how different graph construction methods impact performance? For instance, what performance differences arise when processing reasoning steps as homogeneous graphs vs heterogeneous graphs?
>
> - We thank the reviewer for this interesting suggestion. In our main experiments, we adopt a **homogeneous graph formulation**, where **edges** represent reasoning steps or sub-questions and **nodes** correspond to intermediate answers. We would appreciate clarification from the reviewer on how they envision a heterogeneous graph setup in this context. For example, what distinct node types or edge relations would be considered?
>
> > Q2: Did the authors test the impact of different GNN backbones on performance? Based on the code, it appears they only experimented with the GCN model.
>
> - Thank you for this suggestion. In the main experiments, we indeed adopt **a standard 2-layer GCN** as the backbone for GIBS. To assess the dependence on the backbone, we additionally evaluated GraphSAGE, GAT, and GIN on MoreHopQA with Phi-4 under the same training protocol (Table~X). The results show compatible AUROC across backbones, with GCN slightly outperforming the others, suggesting that our IB-based formulation is the main driver of performance and that a simple GCN is already sufficient to capture the required structural patterns.
>
> **We have added the discussion of these results in Appendix A.4.5.**
>
> **Table 2.** Influence of Different GNN Backbones.
>
> | GNN | GCN | GraphSAGE | GAT | GIN |
> | --- | --- | --- | --- | --- |
> | AUROC | 0.6619 | 0.6446 | 0.5806 | 0.5815
>  |
> > Q3: Can the research topic being studied by the author positively impact RL-based LLM post-training? For example, by utilizing the model you propose to estimate the confidence level of the reasoning chain during the RL rollout process, thereby yielding additional benefits.**
>
> - Our proposed framework is particularly well-suited for RL-based post-training (e.g., PPO or Process Supervision). Specifically, the step-wise confidence scores can serve as a dense process reward during the rollout phase. We have explicitly identified this as a promising research direction in our **Conclusion**. We are excited to explore how integrating our confidence estimation into the RL feedback loop can improve sample efficiency and reasoning robustness. We thank the reviewer for highlighting this high-impact application.

---

### Official Review · Reviewer_m4v1 · 2025-11-01

**Soundness:** 2
**Presentation:** 3
**Contribution:** 2
**Rating:** 4
**Confidence:** 3

**Summary:**

This paper proposes a step-wise confidence attribution (SCA) method for diagnosing large language model reasoning traces. The key idea is to identify consensus steps derived from correct solutions and measure how well individual steps in a given reasoning path align with these consensus steps. The authors present two methods: a non-parametric overlap between reasoning step and consensus steps and a graph-based method which represents reasoning traces as graphs and learns to select subgraphs that align with consensus reasoning graph. Experiments on mathematical reasoning and multi-hop QA datasets demonstrate that the approach can identify erroneous steps and improve final-answer accuracy through error correction.

**Strengths:**

- Step-wise confidence attribution addresses a critical need in making LLM reasoning more interpretable and reliable. The ability to localize where reasoning fails is valuable for model analysis.
- The idea of using consensus steps from correct solutions as anchors for confidence estimation is intuitive and reasonable.

**Weaknesses:**

- The method assumes access to groundtruth answers to construct consensus steps. While this is acceptable for diagnostic evaluation, it undermines claims about error correction. Most baselines in Table 1 do not rely on such supervision, making comparisons somewhat unfair. In Section 5.3, the correction setting implicitly assumes oracle access to correctness feedback. This limits the practicality of the proposed use cases.
- The paper associates "confidence" with "contribution to correct answer," which are 2 distinct concepts. A reasoning step can be confident but incorrect (high model certainty, wrong conclusion). Conversely, a correct step might show low confidence if it's unusual or creative. The method assigns scores based on alignment with consensus from correct trajectories, which measures attribution to correctness rather than confidence.

**Questions:**

1. Why restrict consensus anchors to correct solutions? Have you attempted using consensus from all sampled trajectories rather than correct-only? Could frequent patterns across both correct and incorrect solutions provide better signal?
2. How sensitive is the confidence attribution to the number and diversity of sampled trajectories used for consensus construction?
3. Can the proposed framework operate without ground-truth correctness labels such as self-consistency?

---

> ### Author Response · Authors · 2025-11-28
> **Response to Reviewer m4v1: W1**
>
> We sincerely thank the reviewer for the time and constructive feedback. We are glad that you view step-wise confidence attribution as an important step toward more interpretable and reliable LLM reasoning, and that you find our use of consensus steps from correct solutions as anchors intuitive and reasonable. Below we provide clarifications on your concerns.
>
> > W1 (a): The method assumes access to ground truth answers to construct consensus steps. While this is acceptable for diagnostic evaluation, it undermines claims about error correction. Most baselines in Table 1 do not rely on such supervision, making comparisons somewhat unfair.
>
> We thank the reviewer for raising this important point.
>
> - As the reviewer notes, access to ground-truth answers is **acceptable in diagnostic evaluations**. This is precisely the motivation of our work: the goal is **not** to correct the model in deployment, but to **identify where the reasoning process fails**. For this purpose, the final answer label is both intuitive and fundamental, since diagnostic evaluation requires distinguishing between correct and incorrect reasoning trajectories.
>
> - Importantly, while GIBS uses ground-truth answers **only during training** to construct consensus steps, **no correctness information is required at test time**. At inference time, GIBS takes a single reasoning trace as input. This can be seen in Section 5.5, where GIBS demonstrates strong OOD generalization despite having no access to correctness labels or consensus graphs at test time.
>
> - For white-box methods, the final answer label alone is too **coarse** for stepwise diagnosis. They do not differentiate among individual reasoning steps, but assign confidence based on internal signals such as token log-probabilities, which remain largely **unchanged regardless of whether a trajectory is correct or incorrect.** As a result, the ranking of stepwise confidences within a trajectory stays the same, leading to no improvement in AUROC under evaluation. This is also reflected in Table 1, where white-box methods perform poorly compared to NIBS/GIBS, underscoring the need for fine-grained adjustment of stepwise confidence.
> - Regarding concerns about fairness in comparison, we emphasize that NIBS/GIBS and white-box baselines belong to **different uncertainty-quantification paradigms** rather than competing under identical assumptions. White-box methods rely on internal model states (e.g., token-level likelihoods), whereas our method is a **black-box** approach based on sampling-derived uncertainty signals. Since these paradigms access different sources of information, the comparison highlights complementary strengths and limitations rather than unfair advantage.
>
> - In addition, **as discussed in our response to Q3,** we also evaluate a variant that replaces final-answer correctness labels with self-consistency–based pseudo labels, and observe comparable performance, further demonstrating that our black-box framework remains effective and robust.
>
> > W1(b): In Section 5.3, the correction setting implicitly assumes oracle access to correctness feedback. This limits the practicality of the proposed use cases.
>
> - We appreciate the reviewer’s observation. Section 5.3 studies **an error-correction setting** rather than fully unsupervised self-repair, and in this setting correctness feedback is indeed assumed: the goal is not only to decide whether the final answer is wrong, but also to **identify where the reasoning fails** so that regeneration can focus on the faulty steps. As illustrated in our **case study in Appendix~A.5**, simply telling the LLM that the final answer is incorrect leads to limited gains, whereas providing localized step-wise feedback based on our confidence scores yields more targeted corrections and higher final accuracy.
> - We acknowledge that such oracle feedback may be unavailable in some real-world scenarios. Importantly, however, **GIBS does not rely on final-answer labels at test time:** it takes a single reasoning trace as input and produces step-wise confidence scores in a black-box manner. Thus, even when ground-truth answers are unknown, our method remains applicable as a diagnostic tool.

---

> ### Author Response · Authors · 2025-11-28
> **Response to Reviewer m4v1: W2 and Q1**
>
> > W2:The paper associates "confidence" with "contribution to correct answer," which are 2 distinct concepts. A reasoning step can be confident but incorrect (high model certainty, wrong conclusion). Conversely, a correct step might show low confidence if it's unusual or creative. The method assigns scores based on alignment with consensus from correct trajectories, which measures attribution to correctness rather than confidence.**
>
> - Thank you for the thoughtful comment. We agree that “confidence” and “contribution to correctness” are **conceptually distinct in the black-box setting** of UQ and CE research [1,2,3]. In UQ and CE research [1,2,3], confidence in the black-box setting is typically defined through consistency, such as semantic or behavioral variance. Under this view, **an ideal CE method should assign higher confidence to correct reasoning paths and lower confidence to incorrect ones**, enabling effective evaluation using AUROC, AUPRC, and ACC@C% as well as selective rejection in downstream tasks. Our method follows this paradigm: we define step confidence as “consistency with valid reasoning trajectories,” which directly targets the objective of identifying reliable reasoning steps rather than measuring internal model certainty. This formulation is also designed to address the reviewer’s concern. By aggregating **multiple correct trajectories rather than relying on a single gold solution**, a creative or uncommon but valid step has a much higher chance of appearing in at least one correct trajectory. Then, such steps can still receive non-trivial confidence scores rather than being systematically treated as low-confidence noise.
>
> - Crucially, if we compute the MCS over *all* trajectories without distinguishing between correct and incorrect ones, the confidence estimation becomes ambiguous, since the method no longer knows which distribution to align with. **As shown in Q1, when all trajectories are included (without correctness filtering), the AUROC drops to around 0.5.** This empirical confirmation confirms that correctness-aware alignment is essential for meaningful confidence estimation in this setting.
>
> - We have included this distinction in **Section 3, Problem 2** of the revised version by explicitly separating: **(1) model certainty** (likelihood-based internal confidence), and **(2) correctness-oriented confidence** (consistency with valid reasoning trajectories), and explaining why the latter is the appropriate signal for diagnosing stepwise reasoning reliability.
>
> [1]Geng, Jiahui, et al. "A survey of confidence estimation and calibration in large language models." *Proceedings of the 2024 Conference of the North American Chapter of the Association for Computational Linguistics: Human Language Technologies (Volume 1: Long Papers)*. 2024.
>
> [2]Xiong, Miao, et al. "Can LLMs Express Their Uncertainty? An Empirical Evaluation of Confidence Elicitation in LLMs." *The Twelfth International Conference on Learning Representations*.
>
> [3]Huang, Zhiqi, et al. "Confidence-Based Response Abstinence: Improving LLM Trustworthiness via Activation-Based Uncertainty Estimation." *Proceedings of the 2nd Workshop on Uncertainty-Aware NLP (UncertaiNLP 2025)*. 2025.
>
> > Q1: Why restrict consensus anchors to correct solutions? Have you attempted using consensus from all sampled trajectories rather than correct-only? Could frequent patterns across both correct and incorrect solutions provide better signal?
>
> - We thank the reviewer for the insightful question. We intentionally construct consensus anchors **only from correct trajectories**, as including incorrect ones introduces **systematic bias**. Large language models often exhibit **repeated error patterns or “popular hallucinations”**, where the same incorrect reasoning steps appear across multiple samples. If these are included in the consensus, such erroneous steps are mistakenly treated as high-confidence anchors, blurring the distinction between valid and flawed reasoning.
>
> - To fully address this concern, we additionally conducted an ablation study that uses **all sampled trajectories (both correct and incorrect)** to compute consensus anchors. As shown in **Table 1**, this variant leads to **clear performance degradation**, confirming that mixing incorrect trajectories makes it difficult to separate correct from faulty reasoning paths.
>
> **We have added the discussion of these results in Section 5.5.2.**
>
> **Table 1:** Comparison between all trajectories and Correct Trajectories only.
>
> |  | llama3.1-8b | Deepseek | Phi4 |
> | --- | --- | --- | --- |
> | Correct-only | 0.6471 | 0.8084 | 0.6619 |
> | All trajectories | 0.5192 | 0.6479 | 0.5546 |

---

> ### Author Response · Authors · 2025-11-28
> **Response to Reviewer m4v1: Q2 and Q3**
>
> > Q2: How sensitive is the confidence attribution to the number and diversity of sampled trajectories used for consensus construction?
>
> - To alleviate the reviewer’s concern, we conducted **a sensitivity analysis** to evaluate how the number and diversity of sampled trajectories (N) used for consensus construction affect performance. **The results (see Table 2) show that performance improves as N increases and stabilizes once sufficient diversity is reached (approximately N > 15).** This trend indicates that the consensus becomes more reliable with richer sampling but remains robust beyond a certain threshold.
>
> - In principle, black-box confidence estimation methods approximate the full output distribution by sampling multiple trajectories; hence, performance naturally scales with N before plateauing, which is a behavior consistently observed in prior studies [1,2]. **Following these works and our own empirical analysis, we set N = 20 in all main experiments, which provides a good balance between computational efficiency and stability.**
>
> **We have added this sensitivity analysis in Appendix A.4.5.**
>
> **Table 2.** Sensitivity of SCA to the Number of Sampled Trajectories (MoreHopQA  Phi4).
>
> | # of sample | 5 | 10 | 15 | 20 | 25 |
> | --- | --- | --- | --- | --- | --- |
> | AUROC | 0.5964 | 0.6024 | 0.6569 | 0.6619 | 0.6679 |
>
> [1] Kuhn, Lorenz, Yarin Gal, and Sebastian Farquhar. "Semantic Uncertainty: Linguistic Invariances for Uncertainty Estimation in Natural Language Generation." *The Eleventh International Conference on Learning Representations, 2023*
>
> [2] Lin, Zhen, Shubhendu Trivedi, and Jimeng Sun. "Generating with Confidence: Uncertainty Quantification for Black-box Large Language Models." *Transactions on Machine Learning Research*.
>
> > Q3: Can the proposed framework operate without ground-truth correctness labels such as self-consistency?
>
> Thanks for this insight.
>
> - **[Weakly Supervised Setting]:** Yes. Our framework can operate without explicit gold-answer labels because it is grounded in the **Information Bottleneck (IB) principle**, which **does not** inherently require supervised correctness signals. In **unsupervised or weakly supervised settings**, gold labels can be replaced by **self-consistency–based agreement** among sampled trajectories, where consensus anchors are constructed from trajectories selected via **majority voting**.
>
> - **[New Experiments]**: Our additional experiments show that the effectiveness of this variant depends on how reliably self-consistency identifies correct traces. On **MoreHopQA**, the self-consistent trajectories overlap with the gold-correct set by **≈80%**, resulting in only a small performance drop. In contrast, for **Llama3-8B**, the overlap is only **≈28%**, and performance degrades correspondingly. This pattern indicates that when self-consistency produces sufficiently accurate reference trajectories, our method remains effective without gold labels.
>
> - **[Robustness]**: This analysis further demonstrates the robustness of our approach: even without explicit final-answer labels, our method can leverage self-consistency or other weak supervision signals to achieve comparable performance.
>
> **We have added the discussion of these results in Section 5.5.2.**
>
> **Table 2:** Performance under semi-supervision vs. self-consistency
>
> |  |  | Llama3-8b | Deepseek | Phi4 |
> | --- | --- | --- | --- | --- |
> | MoreHopQA | Correct-only | 0.6471 | 0.8084 | 0.6619 |
> |  | Self-consistency | 0.5734 | 0.7843 | 0.6610 |

---

### Official Review · Reviewer_cv3a · 2025-11-05

**Soundness:** 3
**Presentation:** 3
**Contribution:** 2
**Rating:** 6
**Confidence:** 3

**Summary:**

The paper introduces a framework for diagnosing reasoning errors in large language models without requiring white-box access. The authors propose a Stepwise Confidence Attribution (SCA) framework, which assigns confidence scores to individual reasoning steps using only generated traces and final correctness labels. Specifically, two implementations are presented: NIBS, a non-parametric overlap-based method, and GIBS, a graph-based model leveraging the Information Bottleneck (IB) principle for structure-aware confidence attribution. Experiments across reasoning datasets (GSM8K, Math, MoreHopQA) show that GIBS outperforms baselines and improves reasoning accuracy. The framework also enables targeted self-correction and exhibits out-of-distribution robustness.

**Strengths:**

1. Exploring step-wise attribution frameworks for black-box models is a challenging task, and integrating information theory appears promising.

2. Cross-domain generalization experiments verified that the proposed method possesses a certain degree of scalability.

**Weaknesses:**

1. I think the paper's fundamental assumption that treating common steps as anchors is not fully convincing. From an entropy perspective, these anchors contain less information. A more effective attribution strategy should focus on identifying the correctness of non-consensus steps rather than treating them uniformly.

2. In my view, the compared baselines are insufficient. LLMs used as judges can also assess the correctness of reasoning steps without requiring final-answer labels. Moreover, the utility of the proposed method is not particularly compelling, its advantages over existing techniques are unclear.

3. Lacks a formal analysis of computational complexity, and the computational cost of consensus construction and MCS operations remains high, limiting scalability.

4. Reproducibility requires further clarification, as the paper lacks detailed descriptions of consensus graph construction, model parameter settings, and training procedures.

**Questions:**

please address weaknesses.

---

> ### Author Response · Authors · 2025-11-28
> **Response to Reviewer cv3a: W1 and W2**
>
> We sincerely thank the reviewer for the time and constructive feedback. We are encouraged that you view step-wise attribution for black-box models as a challenging and important direction, find our information-theoretic formulation promising, and appreciate the cross-domain generalization experiments demonstrating scalability. Below, we provide clarifications on your concerns.
>
> > W1: I think the paper's fundamental assumption that treating common steps as anchors is not fully convincing. From an entropy perspective, these anchors contain less information. A more effective attribution strategy should focus on identifying the correctness of non-consensus steps rather than treating them uniformly.
>
> - **[Anchor Information vs. Relevance]** Our assumption is grounded in the Information Bottleneck (IB) framework: the goal is to compress non-essential, high-entropy variations across trajectories while preserving information that is most relevant to correctness I(Z;Y). Intuitively, correct solutions tend to converge on a shared backbone of logic, whereas hallucinated or spurious paths are more diverse. **The consensus part is low-entropy across different reasoning traces, but it is highly informative about the final response. It does not increase the entropy of the trajectory distribution, but instead captures the most stable and reliable signal of correctness.**
>
> - **[Non-Consensus Steps Are Not Treated Uniformly]**: Our framework does **not assign a uniform score to all non-consensus steps**.
>     - In NIBS, Eq.~(2) assigns each step $t_{ij}$ a continuous confidence score based on its degree of semantic alignment with consensus steps, not a binary match. Steps that partially overlap with the consensus structure receive intermediate confidence values.
>     - In GIBS, Eq.~(3) aligns the predicted mask $\mathbf{p}_\theta$ with the consensus graph $G^{MC}$ obtained from MCS matches. Steps that are not part of the consensus graph are not simply suppressed; they are assigned probabilities according to the model’s learned structural alignment. This means our framework already distinguishes degrees of deviation rather than grouping all non-consensus steps together.
>
> We also acknowledge that **non-consensus regions may contain additional useful signals**, and we do not yet explicitly model their internal structure. Designing richer attribution mechanisms for these steps is an important direction for future work, which we now discuss **in the Conclusion and Future Work sections of the revised paper.**
>
> > W2 (a) In my view, the compared baselines are insufficient. LLMs used as judges can also assess the correctness of reasoning steps without requiring final-answer labels.
>
> - **[On LLM-as-Judge Self-Evaluation]** We clarify that we did evaluate an “LLM-as-judge” self-assessment approach through our **P(True)** baseline, where the model predicts step correctness without access to final-answer labels. This baseline yields an AUROC of only **~0.5**, indicating that the model cannot reliably distinguish its own errors due to inherent **overconfidence**.
>
> > W2 (b) Moreover, the utility of the proposed method is not particularly compelling; its advantages over existing techniques are unclear.
>
> We appreciate the reviewer’s feedback and would like to clarify the key advantages and utility of our proposed framework.
>
> - **[Black-box applicability]:** In contrast to white-box confidence methods that depend on internal signals (e.g., token logits or entropy), our framework operates entirely in a black-box setting and **requires only model outputs**. This makes it directly applicable to **closed-source LLMs**, where internal probabilities are unavailable, an increasingly common practical scenario.
>
> - **[Fine-grained interpretability].** Beyond final-answer confidence, our method provides **step-level confidence attribution**, allowing users to pinpoint where the reasoning process fails. As shown in Section 5.3, leveraging step-level confidence for selective correction yields an **improvement of up to 12.3%**, demonstrating clear practical benefits. The case study in Figure 6 further illustrates that identifying the root error (e.g., at Step 1) is crucial for correcting reasoning failures that answer-level signals alone cannot address.

---

> ### Author Response · Authors · 2025-11-28
> **Response to Reviewer cv3a: W3  and W4**
>
> > W3: Lacks a formal analysis of computational complexity, and the computational cost of consensus construction and MCS operations remains high, limiting scalability.
>
> - We clarify that MCS and consensus construction are **one-time offline training costs and are not required during inference.** At test time, GIBS operates as a standard predictive model requiring just **a single forward pass**. As shown in Table 4 (Section 5.4), our method is **three orders of magnitude faster** than test-time MCS approaches while achieving **comparable** performance, demonstrating that it is both scalable and computationally efficient for deployment.
>
> - We analyze the time and space complexity of the proposed MCS algorithm.
> Let $G_1 = (V_1, E_1)$ and $G_2 = (V_2, E_2)$ be two input graphs, and write
> $m_1 = |E_1|$, $m_2 = |E_2|$, and $\mathcal{T}_{\text{NLI}}$ for the time of a single
> forward pass of the NLI model.
>
> - The overall running time of our heuristic MCS algorithm is dominated by pairwise edge–edge similarity computation and is $O(|E_1||E_2| \cdot \mathcal{T}_\text{NLI})$.
>
> - The space complexity is $\mathcal{O}_{\text{space}} = O\bigl(|E_1| |E_2|\bigr).$
>
> A detailed step-by-step complexity analysis of the algorithm is provided **in Appendix A.3** *Complexity Analysis of the Heuristic MCS Algorithm*.
>
> > W4: Reproducibility requires further clarification, as the paper lacks detailed descriptions of consensus graph construction, model parameter settings, and training procedures.**
>
> - **[Reproducibility]** In the revised version, we have significantly expanded the implementation and training details: we now specify the exact LLM backbones and prompting templates (Appendix A.4.2), the construction of consensus graphs via structured generation and rule-based parsing, all similarity backbones and thresholds used in NIBS/GIBS, as well as the GIBS training setup (data scale, GNN architecture, optimization hyperparameters, and evaluation protocol). We also provide a link in our paper to our codebase, further facilitating reproducibility.

---

### Official Review · Reviewer_DoV7 · 2025-11-07

**Soundness:** 3
**Presentation:** 3
**Contribution:** 2
**Rating:** 6
**Confidence:** 4

**Summary:**

The paper Introduces a black-box method to detect where reasoning fails in large language models by assigning confidence scores to each step of a solution.

The author's working hypothesis: correct solutions share common reasoning structures, and steps that deviate from this consensus are likely to be errors.

Proposed approach: two methods: NIBS, which uses semantic similarity, and GIBS, which models reasoning as a graph and learns structural alignment using the Information Bottleneck principle.

**Strengths:**

The black-box error diagnosis framework is something I can resonate with. Also the method is mostly reference free. The authors construct a proxy reference solution (reasoning trace). The application of the IB framework is also largely novel, from what I can tell.

With that being said, I have some concerns. Please see weaknesses.

**Weaknesses:**

1. This "shared structure" hypothesis might not hold true across all domains. Think of a problem which is more creative, and does not follow a deductive reasoning like structure (the GIBS framework largely depends on structure of reasoning). I don’t see this hypothesis playing out there.

2. Line 139. "We begin with the notion of answer-level..." seems like an incomplete sentence ?

3. The semantic similarity as explained in NIBS is already explored in [1] and [2]. The graph structure as explained in GIBS is also partially explored in [3].

4. Does the framework deal with the reasoning error that happen after the first wrong step ? This has not been explicitly mentioned. I would like to know the error identification accuracy of the first wrong reasoning step. the latter steps could skew the accuracy numbers.



[1] ROSCOE: A Suite of Metrics for Scoring Step-by-Step Reasoning

[2] RECEVAL: Evaluating Reasoning Chains via Correctness and Informativeness

[3] Premise-Augmented Reasoning Chains Improve Error Identification in Math reasoning with LLMs

**Questions:**

1. How do you use GSM, MATH for step level error eval ? Do you construct a proxy dataset here ?
2. Can you share results on PRM800K or process bench ?  ( even a sampled subset should be fine )

---

> ### Author Response · Authors · 2025-11-28
> **Response to Reviewer DoV7: W1 -- W4**
>
> We sincerely thank the reviewer for the time and constructive feedback. We are encouraged that you resonate with our black-box, mostly reference-free error diagnosis framework and find our use of the IB principle novel. Below we provide clarifications on your concerns.
>
> >W1: This "shared structure" hypothesis might not hold true across all domains. Think of a problem which is more creative, and does not follow a deductive reasoning like structure (the GIBS framework largely depends on structure of reasoning).
>
> - **[No deductive structure]:** Our framework supports both structured and unstructured reasoning. GIBS applies graph-based alignment when deductive dependencies are present, whereas **NIBS** serves as a **complementary** non-parametric method that **requires no structural assumptions** and measures semantic consistency directly across solutions, making it suitable for less formally organized domains.
>
> - **[Creative domain]:** From the **IB perspective,** our framework aims to **preserve the information most relevant** for identifying valid reasoning trajectories. In creative domains such as MorehopQA, surface forms can vary more than in structured tasks like math, but valid reasoning paths still **exhibit latent semantic regularities** that reflect the underlying logic of the solution. By capturing these regularities, our method can identify steps that deviate from the valid trajectory and flag them as **inconsistencies**.
>
> > W2: Line 139. seems like an incomplete sentence ?
>
> - Thank you for catching this. We have corrected the sentence to read: “We begin with the notion of answer-level confidence estimation (CE).”
>
> > W3: The semantic similarity as explained in NIBS is already explored in [1] and [2]. The graph structure as explained in GIBS is also partially explored in [3].
>
> - We thank the reviewer for pointing out these works. Although our methods share the high-level goal of localizing errors, the setting and mechanisms differ: [1,2,3] operate on a single reasoning chain (often with whole-chain or binary step labels), whereas NIBS/GIBS exploit multi-sample consistency across multiple trajectories and produce step-wise confidence scores built from multiple sampled solutions.
>
> - To directly address this concern, we adapt ROSCOE-ss/sa and PARC to our setting and evaluate them as baselines; they perform worse than NIBS/GIBS across AUROC, AUCPR, and ACC@80% on MultiHopQA with Llama3.1-8B, DeepSeek, and Phi-4 (see Table 1).
>
> **We have also added an citation of [1–3] in the related work section.**
>
> **Table 1**: Performance of NIBS/GIBS and adapted ROSCOE-ss/sa and PARC baselines.
>
> |  | Llama3.1-8b |  |  | Deepseek |  |  | Phi4 |  |  |
> | --- | --- | --- | --- | --- | --- | --- | --- | --- | --- |
> |  | AUROC | AUCPR | ACC@80% | AUROC | AUCPR | ACC@80% | AUROC | AUCPR | ACC@80% |
> | ROSCOE-ss | 0.477 | 0.517 | 0.503 | 0.408 | 0.584 | 0.528 | 0.421 | 0.603 | 0.542 |
> | ROSCOE-sa | 0.524 | 0.565 | 0.541 | 0.492 | 0.600 | 0.612 | 0.488 | 0.594 | 0.613 |
> | PARC | 0.589 | 0.608 | 0.511 | 0.637 | 0.686 | 0.659 | 0.606 | 0.676 | 0.650 |
> | NIBS | 0.512 | 0.553 | 0.531 | 0.666 | 0.776 | 0.649 | 0.580 | 0.663 | 0.696 |
> | GIBS | **0.647** | **0.669** | **0.560** | **0.808** | **0.835** | **0.705** | **0.661** | **0.686** | **0.705** |
>
> - **W4:** Does the framework deal with the reasoning error that happen after the first wrong step ? I would like to know the error identification accuracy of the first wrong reasoning step. the latter steps could skew the accuracy numbers.
>
> - **[New Experiments]:** We clarify that our framework **targets all erroneous steps**, as confidence attribution is computed at every intermediate stage. To isolate performance on the earliest mistake and avoid the effects of error propagation, we conducted an additional experiment **evaluating only the first error step**. Table 2 reports AUROC and Adaptive F1 (best F1 across thresholds) **on MoreHopQA with Phi4**. The results show that GIBS consistently **outperforms the baselines** and is highly sensitive to the initial divergence point. These results will be added to **Appendix A.6**.
>
> **Table 2.** First Error Step Detection Performance (AUROC and Adaptive F1) on MoreHopQA.
> |  | Llama3.1-8b |  | Deepseek |  | Phi4 |  |
> | --- | --- | --- | --- | --- | --- | --- |
> |  | AUROC | F1 | AUROC | F1 | AUROC | F1 |
> | P(true) | 0.5126 | **0.8400** | 0.5152 | 0.8518 | 0.5090 | 0.8750 |
> | SL(norm) | 0.4941 | 0.8376 | 0.3009 | 0.8518 | 0.4724 | 0.8749 |
> | Entropy | 0.5570 | 0.8376 | 0.6004 | 0.8518 | 0.6837 | 0.8748 |
> | LECO | 0.3839 | 0.8363 | 0.3307 | 0.8332 | 0.3884 | 0.8723 |
> | Cos-Max | 0.4275 | 0.8366 | 0.4529 | 0.8655 | 0.4772 | 0.8871 |
> | Cos-Mean | 0.4863 | 0.8366 | 0.5313 | 0.8655 | 0.5099 | 0.8871 |
> | NLI-Max | 0.5375 | 0.8366 | 0.7110 | 0.8655 | 0.6632 | **0.8883** |
> | NLI-Mean | 0.5285 | 0.8366 | 0.6544 | 0.8655 | 0.5824 | 0.8868 |
> | GIBS | **0.6164** | 0.8376 | **0.7885** | **0.8681** | **0.6841** | 0.8786 |

---

> ### Author Response · Authors · 2025-11-28
> **Response to Reviewer DoV7: Q1 and Q2**
>
> >  Q1: How do you use GSM, MATH for step-level error eval ? Do you construct a proxy dataset here ?
>
> - **[Step-Level Evaluation Protocol]:** We do **not** construct any proxy dataset. For each dataset, we evaluate using multiple sampled trajectories from different LLMs. As described in Section 4.4 and Appendix A.4.4, we follow prior work [1,2,3,4] by using an LLM-as-a-Judge approach (e.g., GPT-4o) to annotate step-wise correctness. These labels are used solely for evaluation and are not used for **training the model**.
>
> > Q2: Can you share results on PRM800K or process bench ? ( even a sampled subset should be fine )
>
> - Thanks for the feedback and we have included results on the PRM800K dataset (released by OpenAI)[5], **which contains unstructured, free-form GPT-4 traces with high-quality human step-level annotations.**
>
> **(1) Experimental Setup:** To ensure distributional consistency, we sample **N = 20 diverse reasoning paths per question** directly from the dataset. Each step is evaluated using the **gold human-provided labels**. For GIBS, we represent each CoT as a **linear graph**: each sentence is treated as an **edge**, nodes are left empty, and edges are connected sequentially to reflect the step-by-step progression. We follow the same training protocol as in our main experiments, using 2k reasoning paths for training and 10k for testing.
>
> **(2) Results & Advantage:** Table 3 reports the results. Because PRM800K is based on **pre-collected GPT-4 traces**, it reflects a **black-box** setting where white-box baselines, which require token log-probabilities, are not applicable. In contrast, our methods (NIBS and GIBS) function effectively in this setting. As shown in Table 3, both **NIBS and GIBS perform well** under this setting. The gap mainly reflects that PRM800K provides ***limited structural information*,** so GIBS cannot fully leverage the richer dependencies it uses with structured outputs. When such a structure is available, GIBS will benefit more.
>
> **We have updated the PRM800K results in Section 5.5.3 as part of our generalization analysis.**
>
> **Table 3.** Performance of NIBS and GIBS on the PRM800K dataset.
>
> |  | AUROC | AUCPR | ACC@80% |
> | --- | --- | --- | --- |
> | White-box methods | N/A | N/A | N/A |
> | Cos-Max | 0.6156  | 0.7840 | 0.7734 |
> | Cos-Mean | 0.6821 | 0.8343 | 0.7860 |
> | NLI-Max | 0.7666 | 0.9074 | 0.7899  |
> | NLI-Mean | 0.8181 | 0.9019 | 0.8573 |
> | GIBS | 0.6556 | 0.8203 | 0.7570 |
>
> [1] Ye, Zihuiwen, et al. "Uncertainty-Aware Step-wise Verification with Generative Reward Models." *ICLR Workshop: Quantify Uncertainty and Hallucination in Foundation Models: The Next Frontier in Reliable AI*.
>
> [2]Lin, Zhen, Shubhendu Trivedi, and Jimeng Sun. "Generating with Confidence: Uncertainty Quantification for Black-box Large Language Models." *Transactions on Machine Learning Research*.
>
> [3]Xiong, Miao, et al. "Can LLMs Express Their Uncertainty? An Empirical Evaluation of Confidence Elicitation in LLMs." *The Twelfth International Conference on Learning Representations*.
>
> [4]Huang, Zhiqi, et al. "Confidence-Based Response Abstinence: Improving LLM Trustworthiness via Activation-Based Uncertainty Estimation." *Proceedings of the 2nd Workshop on Uncertainty-Aware NLP (UncertaiNLP 2025)*. 2025.
>
> [5] Lightman, Hunter, et al. "Let's verify step by step." *The Twelfth International Conference on Learning Representations*. 2023.

---

### Meta-Review · Area_Chair_myx3 · 2026-01-05

**Summary:**

This work proposes a Stepwise Confidence Attribution method. Under the premise of only providing reasoning trajectories, it constructs reasoning graphs and uses information bottleneck to assign confidence to nodes within these graphs, thereby achieving consensus-based step-level confidence attribution.

Reviewers agree that attributing errors to specific steps is a critical and valuable research direction, and they find the method novel with strong generalization capabilities.

The reviewers raised the following concerns:

1. Reasonableness and Limitations of the Core Assumption: The method is based on the assumption that "correct solutions share a common reasoning structure." Several reviewers questioned whether this assumption holds in more creative or non-deductive reasoning domains, such as open-ended generation.

2. Lack of Clarity in Problem Definition, Evaluation Setup, and Experimental Details.

3. Strong Dependence on Multiple External Models: The method heavily relies on several external models, and a failure in any one of them could compromise the results.

**Reviewer Concerns:**

In response, the author improved the clarity of the paper. However, the reasonableness of the core assumption and the robustness of the method remain significant risks.

**Reviewer Scores:**

After thorough discussion, reviewer 2s99 would raise their score to 4, while the other reviewers are inclined to keep their scores unchanged.

---

### Decision · Program_Chairs · 2026-01-26

Reject